



# Sea-level rise contribution from Ryder Glacier in Northern Greenland varies by an order of magnitude by 2300 depending on future emissions

Felicity A Holmes[1,2], Jamie Barnett[1,2], Henning Åkesson[3], Mathieu Morlighem[4], Johan Nilsson[2,5], Nina Kirchner[2,6,7], and Martin Jakobsson[1,2]

[1]Department of Geological Sciences, Stockholm University, Sweden
[2]Bolin Centre for Climate Research, Stockholm University, Sweden
[3]Department of Geosciences, University of Oslo, Norway
[4]Department of Earth Sciences, Dartmouth College, USA
[5]Department of Meteorology, Stockholm University, Sweden
[6]Department of Physical Geography, Stockholm University, Sweden
[7]Tarfala Research Station, Stockholm University, Sweden

**Correspondence:** Felicity Holmes (felicity.holmes@geo.su.se)

**Abstract.** Northern Greenland contains some of the ice sheet's last remaining glaciers with floating ice tongues. One of these glaciers is Ryder Glacier, which has been relatively stable in recent decades in contrast to the neighbouring Petermann and C.H. Ostenfeld Glaciers. Previous research suggests that fjord and bedrock topography, rather than atmospheric or ocean temperatures, may be the main factor behind these differences. Understanding Ryder Glacier's future behaviour is important as ice-tongue loss could lead to acceleration and increased mass loss from discharge. In contrast to Petermann and C.H. Ostenfeld Glaciers, where the impact of ice-tongue loss has been established by several studies, no previous research has assessed the risk and consequences of ice-tongue loss at Ryder Glacier. Meanwhile, it is unclear whether Greenland-wide modelling attempts are able to accurately resolve the influence of fjord/bedrock topography and small scale variations in ice dynamics for a glacier like Ryder. To fill these gaps, we here conduct targeted high-resolution modelling to Ryder Glacier until the year 2300. Thereby we aim to reduce uncertainties in sea-level rise projections from this area. We find that mass loss is dominated by discharge under all scenarios until 2100, after which surface mass balance losses take over under a high emissions future, leading to a much higher sea level rise contribution.

## 1  Introduction

The Greenland Ice Sheet is currently the largest single contributor to global sea-level rise, having exhibited an acceleration of mass loss during the last two decades (Briner et al., 2020; King et al., 2020; Millan et al., 2023; Greene et al., 2024). Mass loss occurs both due to a negative surface mass balance and through dynamic loss directly into the oceans - with these dynamic processes having driven the recent increase in mass loss (King et al., 2020). Current air temperatures in North Greenland are already higher that at any time during the past millennium (Hörhold et al., 2023), and current projections suggest we are on track to experience rates of mass loss that have not been seen during the entire Holocene (Briner et al., 2020; Goelzer et al.,





2020). Paleo-evidence suggests that the Greenland ice sheet as a whole is sensitive to substantial deglaciation in climates that are only slightly warmer than at present (Schaefer et al., 2016), and that a rise in global mean temperatures of around 2°C above pre-industrial temperatures may be enough to trigger self-sustained melting of the Greenland ice sheet (Bochow et al., 2023). However, the large degree of spatial heterogeneity in Greenland outlet glacier behaviour contributes to a large uncertainty in the response of these glaciers to ocean forcing (Mouginot et al., 2019).

The Northern sector of the Greenland Ice Sheet has been relatively understudied (Hill et al., 2018b), with uncertainties over its future behaviour having been found to be higher than those associated with other sectors of the ice sheet (Choi et al., 2021). One reason for is that this this sector contains the last few remaining ice tongues in Greenland (Millan et al., 2023). If these floating ice tongues are lost, the reduction in buttressing has the potential to accelerate upstream ice flow and associated dynamic mass loss (Millan et al., 2023). In total, this sector has the potential to contribute c. 93 cm to global

sea-level rise (Mouginot et al., 2019), with the associated increased freshwater release to the oceans having knock-on impacts through changes on biogeochemical conditions (Kanna et al., 2022) and through potential weakening of the Atlantic Meridional Overturning Circulation (Yang et al., 2016).

Observational records from recent decades have shown a large degree of spatial heterogeneity between neighbouring glaciers, underscoring the high uncertainty surrounding the future behaviour of glaciers. As such, recent research has concluded that

changes need to be considered at the scale of individual glaciers (Cooper et al., 2022), and that seemingly stable glaciers are key targets for ongoing research due to the potential for rapid retreat to occur out-of-sync with climate forcing (Robel et al., 2022). Heterogeneity in glacier behaviour is well exemplified by considering three North Greenlandic glaciers; Petermann , Ryder, and C.H. Ostenfeld glaciers (See Fig. 1). Petermann and Ryder glaciers are two of the few Greenlandic glaciers which still have a floating ice tongue. Petermann Glacier has been retreating, having lost around 40% of its ice tongue over the past 15

years (Münchow et al., 2016), whilst Ryder Glacier has remained reasonably stable. Petermann Glacier exhibited an average terminus retreat rate of 311 m yr⁻¹ between 1948 and 2015 at the same time as Ryder Glacier experienced little overall change in margin position despite periods of both advance and retreat (Hill et al., 2018b; Holmes et al., 2021). Meanwhile, C.H. Ostenfeld glacier lost its c. 20 km long ice tongue between 2002 and 2003 and has since had a relatively stable front position (Hill et al., 2018b). This raises the following questions, which we address in this study: Why have these three neighbouring

glaciers exhibited contrasting ice-tongue behaviour in recent decades, and what does this mean for their future response to a warming climate?

## 2   Petermann, Ryder, and C.H. Ostenfeld Glaciers

The first possible explanation for the different behaviour of Petermann, Ryder, and C.H. Ostenfeld glaciers is differences in their meteorological setting and therefore surface mass balance (SMB). Records of the SMB for each glacier from 1958 to

2018 are available from Mouginot et al. (2019) and show an average SMB of 9.4 Gt yr⁻¹ for Petermann Glacier, 2.8 Gt yr⁻¹ for Ryder Glacier, and 1.5 Gt yr⁻¹ for C.H. Ostenfeld Glacier. Although this at first seems to show large differences between the three glaciers, accounting for the area of each glacier's drainage basin reveals that the differences are small; the SMB in



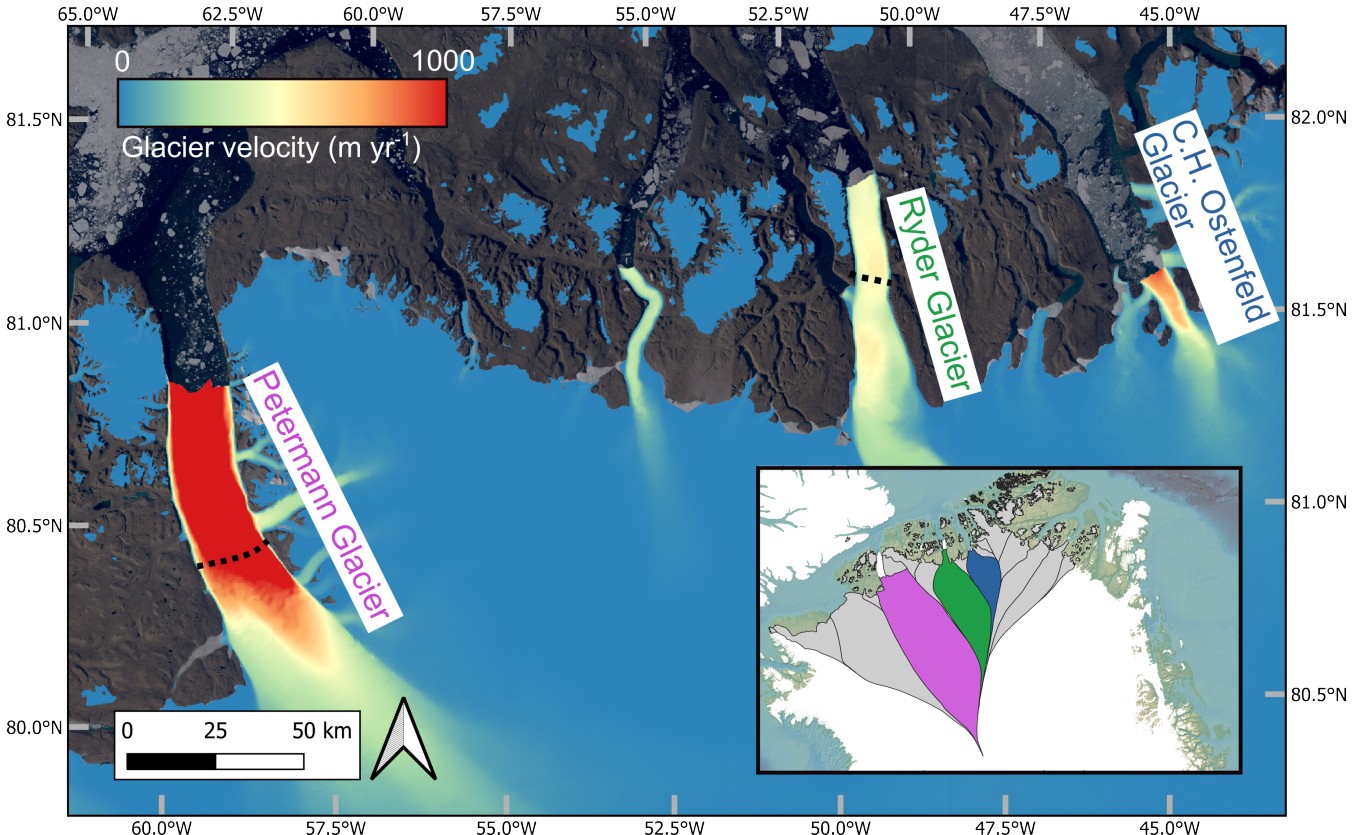

**Figure 1.** Velocities of Petermann, Ryder, and C.H. Ostenfeld glaciers in Northern Greenland. The location of the drainage basins for all Northern Greenlandic glaciers are shown in the inset figure, with the basins for Petermann, Ryder, and C.H. Ostenfeld glaciers coloured to correspond with the text label for each glacier (Mouginot et al., 2019). Approximate grounding line positions are denoted by dotted black lines and are from Millan et al. (2023). The satellite image is a 2015 Sentinel-2 image mosaic from MacGregor et al. (2020). The surface velocities are from 2015-2016 and are sourced from the MEaSUREs dataset (Joughin et al., 2016, 2018).

kilotonnes per $km^2$ is 126.4 for Petermann Glacier, 93.0 for Ryder Glacier, and 124.1 for C.H. Ostenfeld Glacier. In fact, whilst all glaciers have an overall positive SMB, it is Ryder Glacier - which has exhibited a stable front position - that has the lowest
SMB per unit area.

Ocean-driven melting is the next possible explanatory factor, and there have been several studies focusing on basal melt rates at Petermann and Ryder glaciers. It has previously been suggested that a shallow bathymetric sill in front of Ryder Glacier's grounding line may be responsible for its relative stability through blocking intrusions of warm Atlantic water and so keeping melt rates under the ice tongue low (Jakobsson et al., 2020; Nilsson et al., 2023). This is in contrast to Petermann Glacier,
which has a bathymetric sill with a deeper channel shown to allow greater intrusion of warm Atlantic water (Jakobsson et al., 2018). Previous observational records of basal melt show the highest melt rates of nearly 60 m yr$^{-1}$ are found in localised areas at Ryder Glacier, but that large areas of Petermann Glacier's ice tongue experience high melt rates above 40 m yr$^{-1}$ (Wilson





et al., 2017; Rückamp et al., 2019). Recent satellite-derived observations of ice tongue melt at Petermann and Ryder Glaciers have indicated significantly higher grounding line melt rates at Ryder Glacier during the period 2000-2020 (Millan et al.,

2023). Although temporal variations in melt rates at both glaciers tend to follow variations in ocean temperatures, the absolute grounding line melt rates at Ryder Glacier have most often been between 40 and 50 m yr$^{-1}$ compared to only 10 to 20 m yr$^{-1}$ at Petermann Glacier (Millan et al., 2023). Turning our attention to the break up of C.H. Ostenfeld's ice tongue in the early 2000s, the few existing oceanographic measurements on the continental slope off the Lincoln Sea reveal lower Atlantic Water temperatures in the early to mid 1990s followed by warmer temperatures in the early 2000s (De Steur et al., 2013). However,

this Atlantic Water temperature record lacks observations between 1996 and 2003, the period roughly coinciding with C. H. Ostenfeld's collapse. Accordingly, it is difficult to determine whether regionally warmer Atlantic Water temperatures can have contributed to the disintegration of C. H. Ostenfeld's ice tongue. Compounding the uncertainties, the lack of well-constrained bathymetry in Victoria Fjord (for C.H. Ostenfeld Glacier) or under the ice tongues of Ryder and Petermann glaciers means that open questions remain over how much of the warm Atlantic water identified in the region is able to reach all the way to the

grounding line. Thus, despite uncertainties, the available data available suggests that the greatest levels of ocean-melt driven mass loss are found at the glacier with the most stable front position - Ryder Glacier.

This leaves bedrock topography and fjord geometry as two interconnected explanatory variables. The basal topography from BedMachine (Morlighem et al., 2017) shows that the grounding line and front positions of Petermann and C.H. Ostenfeld glaciers are located on mostly flat topography, although the bathymetry below the ice tongues is less well-known than the

subglacial topography. Ryder Glacier's current grounding line is also located on relatively flat topography, but has recently exhibited retreat on its eastern margin (Millan et al., 2023) and has a deep trough around 20 km inland - something which may lead to rapid retreat in the future (Hill et al., 2018b). In terms of fjord geometry, the pre-2002 ice tongue at C.H. Ostenfeld glacier was not attached to the fjord walls as a result of a fjord which widens seaward of its present day terminus position. In contrast, both Petermann and Ryder Glaciers terminate in fjords that narrow towards the glacier termini and show contact

between the fjord walls and ice-tongue edges. This lack of lateral friction at C.H. Ostenfeld's ice tongue could have made it more vulnerable to collapse, whilst the topographically constrained ice tongues of Petermann and Ryder Glacier have survived to the present day. But, if the geometrical setting is similar for Petermann and Ryder Glaciers, why has only Petermann Glacier shown retreat? One possible explanation is that smaller tributary glaciers flow into Petermann Glacier's ice tongue along its lateral boundaries, pushing the ice tongue edges away from the fjord walls. Consideration of satellite imagery shows that cracks

in Petermann Glacier's ice tongue start to develop directly after these tributaries flow into the ice tongue. Although rifts can also be seen at Ryder Glacier, they are not as numerous or clearly defined. When taken in conjunction with the SMB and ocean-driven melt considerations detailed above, it seems that fjord geometry and bedrock topography are likely of great importance for explaining the contrasting histories of these three glaciers. This suggests that these factors will also be of importance in the future, even if rising atmospheric and ocean temperatures control the direction of change.

Looking towards the future, several studies have identified the need to understand the dynamic impact of ice-tongue loss in Northern Greenland, as a result of the knock-on impacts on sea-level rise projections (Millan et al., 2018; Hill et al., 2018b; Millan et al., 2023). Specifically, there is the possibility for discharge losses to increase substantially after the break-up of




an ice tongue due to the loss of buttressing. This behaviour is however complex and varies from glacier to glacier, making its incorporation in future projections an area of uncertainty. This is of particular importance in the present day as recent observations have suggested that all of Greenland's ice tongues have experienced considerable weakening during the last few decades (Millan et al., 2023). Whilst there have been many targeted studies of Petermann Glacier (Nick et al., 2012; Hill et al., 2018a; Åkesson et al., 2022) and analysis of C.H. Ostenfeld Glacier's response to ice-tongue collapse (Hill et al., 2018b), we are not aware of any published studies focusing on Ryder Glacier and how it may respond to future ice-tongue collapse. Whilst several studies have targeted the future behaviour and sea-level rise contribution from the entire Greenland ice sheet, the associated computational cost means that the spatial resolution of these models is often relatively coarse, and processes such as ice-ocean interactions are crudely represented (Aschwanden et al., 2019; Beckmann and Winkelmann, 2023; Choi et al., 2021). These considerations lead to two key research questions: i) How will Ryder Glacier respond to future climate change, and what are the implications for sea-level rise? ii) How does potential future ice-tongue loss at Ryder Glacier compare to neighbouring systems in North Greenland, and what does this tell us about the controls on ice-tongue and glacier evolution?

To answer these questions, we conduct targeted, high-resolution simulations of Ryder Glacier's response to future projected atmospheric and oceanic changes. Ryder Glacier is of particular interest in light of its current and rather puzzling stability and, unlike Petermann Glacier, has not previously been the focus of targeted modelling studies. Through disentangling the controls on glacier evolution under different magnitudes of climate warming, the results from our simulations will have implications for other glacier-fjord systems. Background on the numerical model is given below, followed by a description and discussion of the model results.

## 3  Numerical model

The numerical model employed in this work is the Ice-Sheet and Sea-Level System Model (ISSM) (Larour et al., 2012), a finite-element model that has been used extensively to model glaciers in both Greenland and Antarctica.

The model domain encompasses the entire drainage basin of Ryder Glacier as defined by Mouginot et al. (2019), and is extended up until the mouth of Sherard Osborn fjord to allow for potential glacier advance (Fig. 1). The mesh is extruded to have seven vertical layers, and the horizontal mesh resolution varies between 300 m and 10 000 m based on surface velocity, with areas flowing faster than 500 m yr$^{-1}$ having the finest resolution.

For all simulations, a higher-order (HO) approximation, the 'Blatter-Pattyn' approximation, of ice flow was used, which is computationally cheaper than Full Stokes (Blatter, 1995; Pattyn, 2003). Previous studies have compared this HO approximation to Full-Stokes set-ups and found limited differences, performing considerably better than the even cheaper Shelfy-Stream Approximation (Yu et al., 2018). For transient simulations, the time step size was set to 0.083 years, corresponding to a monthly time step. In scenarios where large accelerations were observed, the simulations were re-run with a timestep of three days (0.008 years) in order to satisfy the Courant–Friedrichs–Lewy condition.



### 3.1 Topographic data

130 For the bathymetry and bedrock topography, we use BedMachine v5 (Morlighem et al., 2017) which includes the newest bathymetric data from the *Ryder 2019* expedition to Sherard Osborn fjord with the Swedish Icebreaker Oden (Jakobsson et al., 2020). For the initial surface elevation we use the GIMP DEM with a nominal date of 2007 (Howat et al., 2014), which is provided along with BedMachine v5 (Morlighem et al., 2017).

### 3.2 Modelling workflow

135 The first step in the workflow was to invert for basal friction under glaciated areas, by minimising the misfit between modelled velocities and observed 1995-2015 velocities from MEaSUREs (Joughin et al., 2016, 2018).

For all simulations, a Budd-type friction law is used. This law was found by a comprehensive comparison of several friction laws to work best for the neighbouring Petermann Glacier (Åkesson et al., 2021). This friction law has additionally been used successfully for simulations covering the entire Greenland ice sheet (Choi et al., 2021). Here, basal drag $\boldsymbol{\tau}_b$ is based on the 140 inverted friction parameter $\alpha$, basal velocity $\mathbf{u}_b$, and the effective pressure $N$, which itself is calculated from ice density $\rho_{\mathrm{i}}$, water density $\rho_{\mathrm{w}}$, gravitational acceleration $g$, ice thickness $H$, and the ice base elevation $b$, as below:

$$\boldsymbol{\tau}_b = -\alpha^2 N \mathbf{u}_b, N = \rho_{\mathrm{i}} g H - \rho_{\mathrm{w}} g b. \tag{1}$$

Similarly to Åkesson et al. (2022), an inversion was also performed to infer a spatially variable rheology of the ice tongue, using the same velocity dataset as used to infer the friction parameter $\alpha$ under grounded ice. The rest of the domain is assumed 145 to have a constant viscosity matching the behaviour of an ice temperature of -12°C.

After initialisation, a relaxation simulation is run for 50 years to both calibrate the calving parameters (see Eq. 2) and validate the model against present-day conditions (Fig. 2). In this relaxation, we use a temporally fixed SMB forcing equal to the mean of 1950-2014 SMB from RACMOv2.3p2 (Noël et al., 2022). Melt below the ice tongue was set to be 40 m yr⁻¹ at water depths equal to or greater than 300 m, linearly decreasing to 0 m yr⁻¹ at depths equal to or shallower than 100 m. These melt rates 150 were chosen as they lie within the range of published present-day melt rate estimates for Ryder Glacier's ice tongue from various observational and modelling studies (Wilson et al., 2017; Wiskandt et al., 2023; Millan et al., 2023). During the 50 year relaxation simulation, Ryder Glacier exhibits mass loss equal to 0.9 Gt yr⁻¹, which is similar to the observed mass loss between 2000 and 2017 of 0.96 Gt yr⁻¹ (Mouginot et al., 2019). The front position remains stable whilst a small grounding line retreat of c. 2 km is seen - corresponding to recent reports of Ryder Glacier's behaviour (Holmes et al., 2021; Millan et al., 2023). 155 Modelled velocities at the end of the initialisation match well with observed velocities, with ice-tongue speeds of around 550 m yr⁻¹ (Fig. 2).



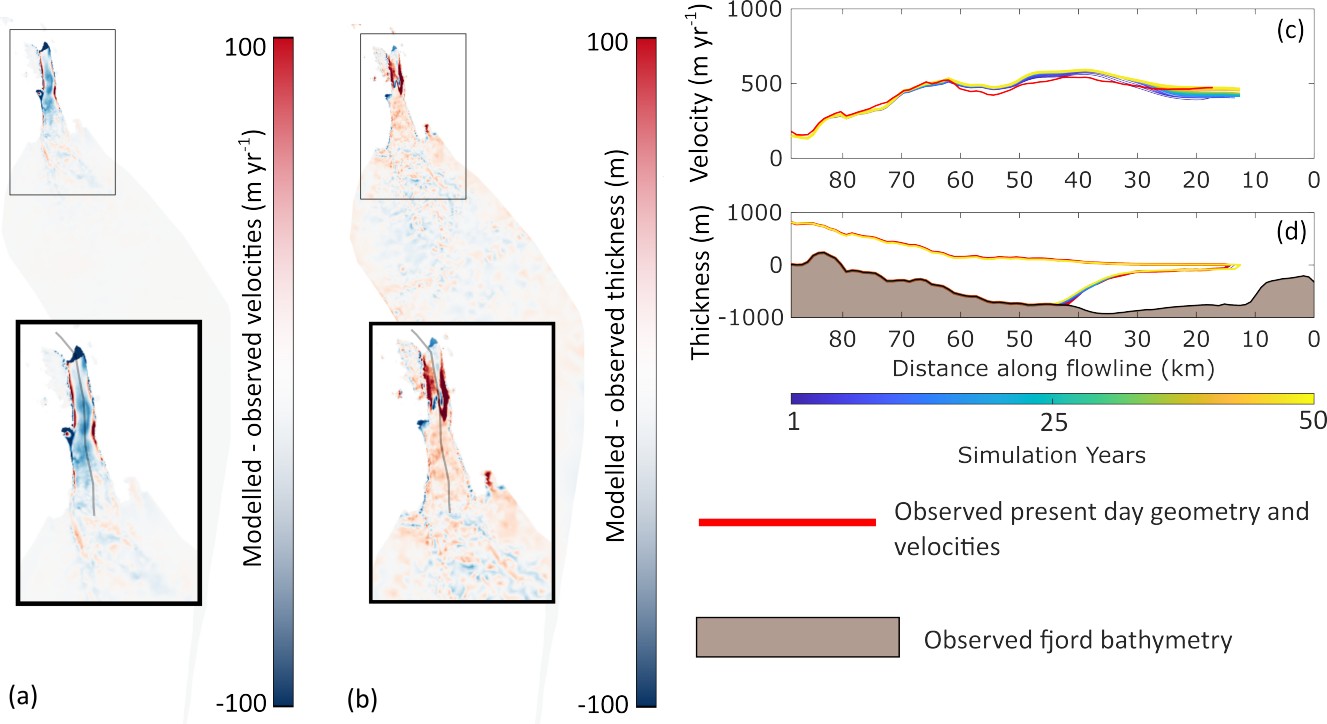

**Figure 2.** Modelled state of Ryder Glacier at the end of the relaxation. The misfit between modelled and observed velocities is shown in (a), alongside the thickness misfits in (b), with a inset close-up of the frontal area shown for both metrics. In (c), the glacier velocity along the central flowline (grey line on the inset panels in (a) and (b)) is shown, with the observed velocities plotted in red. In (d), the glacier geometry along the flowline is shown, with the observed present day geometry in red. Observed thickness and geometry data is from BedMachine v5 Morlighem et al. (2017) and observed velocities are from MEaSUREs Joughin et al. (2016, 2018).

### 3.2.1 Atmospheric-driven forcing

To improve the representation of surface mass balance (SMB) as the ice elevation changes with time, we set a relation between surface elevation and SMB to account for the melt-elevation feedback, which is often neglected in similar studies (Nick et al.,
2013; Bassis et al., 2017; Steiger et al., 2018). This is incorporated through the use of the SMB gradients module implemented in ISSM, where the SMB from a reference dataset is altered each time step to account for changes in surface elevation (Helsen et al., 2012). The gradient method is only applied in the ablation zone (evaluated at each time step based upon where the reference dataset shows a negative SMB), due to uncertainties regarding the melt-elevation relationship in the accumulation zone (Helsen et al., 2012; Calov et al., 2018; Choi et al., 2018). The reference data sets used for this study are various outputs
from RACMOv2.3p2 at a 1 km resolution; the mean SMB from the period 1950-2014 is used for the relaxation, and monthly SMB fields for SSPs 1-2.6 and 5-8.5 are used when running transient simulations to 2300 (Noël et al., 2022). A simulation with no SMB-gradients feedback is also run for both of the emissions scenarios, so that the impact of this feedback can be assessed.



### 3.2.2 Ocean-driven forcing

In the relaxation, basal melting varies linearly with depth in a manner consistent with observations from Northern Greenland
(Slater and Straneo, 2022), and with the maximum applied melt rate based on mean 2011-2015 observed melt rates at Ryder
Glacier from Wilson et al. (2017). However, observations additionally show considerable lateral variation in grounding line
melt rates (e.g. between 10 m yr$^{-1}$ and 60 m yr$^{-1}$) - something that is not accounted for here. For the future transient simulations,
the maximum melt rate is varied to investigate the impact of increased ocean thermal forcing. Subsurface ocean temperatures
around North Greenland are projected to increase by around 1°C by 2100 and around 2°C by 2200 (Yin et al., 2011). Recent
work suggests that the increase in melt rates per each degree ocean warming depends on the magnitude of subglacial discharge
(Slater and Straneo, 2022; Wiskandt et al., 2023). For Ryder Glacier, melt rates are likely to increase by c. 12 m yr$^{-1}$ °C$^{-1}$
in the absence of any subglacial discharge and by c. 20 m yr$^{-1}$ °C$^{-1}$ assuming a high level of subglacial discharge (Wiskandt
et al., 2023). Assuming warm Atlantic waters will be able to reach the grounding line of Ryder Glacier, maximum melt rates at
Ryder Glacier are likely to increase by between 12 m yr$^{-1}$ and 20 m yr$^{-1}$ by 2100 and by 24 to 40 m yr$^{-1}$ by 2200. Assuming the
warming trend continues linearly to 2300, this would mean an additional 36 to 60 m yr$^{-1}$ of melt by 2300. However, this linear
relationship between ocean temperature and melt rate may be on the conservative side; a combined observational–modelling
study at 79N Glacier (another north Greenlandic glacier with an ice tongue) found a quadratic relationship between ocean
temperatures and basal melt rates under the ice tongue (Wekerle et al., 2024). As such, we also impose nonlinear increases
in melt rates for some simulations (see Sect. 3.3). Thus, although we do not explicitly model the impact of supraglacial melt/
subglacial discharge, the suite of different basal melt scenarios emulate the impact of these processes based on present day
observations of the relationships between subglacial discharge and basal melt. In reality, temporal variations on seasonal,
inter-annual, and interdecadal scales also occur (Christian et al., 2020). However, these variations are often stochastic and
not straightforward to explicitly include in simulations of the future. A fully coupled glacier-ocean-atmosphere model would
allow for better representation of these processes, but would require considerable computational resources and more data for
validation.

We use a von Mises calving law, which computes the calving rate $c$ depending on the terminus velocity $\mathbf{u}$ and the von Mises
tensile stress $\tilde{\sigma}$ (Morlighem et al., 2016). In this formulation, calving occurs where the von Mises tensile stress is greater than
the tuned stress threshold parameter $\sigma_{\max}$. This calving law was chosen as it has been found to work well in both Greenland
and Antarctica (Choi et al., 2018; Wilner et al., 2023).

$$c = |\mathbf{u}|\frac{\tilde{\sigma}}{\sigma_{\max}}, \qquad (2)$$

The stress threshold $\sigma_{\max}$ for floating ice is set to 200 kPa. This value allowed the glacier to best match observations of front
position during the 50-year transient relaxation.

Finally, the stress threshold for grounded ice is set so to 500 kPa, as this led to the best match with Ryder Glacier's grounded
calving front during the relaxation.





**Table 1.** Summary of forcings applied in each of the transient simulations to 2300. All low emissions scenarios are abbreviated to 'LE' and all high emissions scenarios to 'HE'. Simulations are also labelled 'NG' where no SMB-elevation feedback (or 'no gradient') is included. The melt scenario imposed is denoted as LMI for a low melt increase, HMI for a high melt increase, and MMI for a maximum melt increase. Where increased calving is simulated through a reduction in the calving stress threshold this is abbreviated to IC. Simulations that increase calving and increase submarine melt rates are given the suffix IC/IM.

| Simulation name | Surface Mass Balance | Floating $\sigma_{\max}$ (kPa) | Grounded $\sigma_{\max}$ (kPa) | Max. basal melt rate (m/a) |
|---|---|---|---|---|
| LE | SSP1-2.6 | 200 | 500 | 40 |
| HE | SSP5-8.5 | 200 | 500 | 40 |
| LE$_{NG}$ | SSP1-2.6 (no gradient) | 200 | 500 | 40 |
| HE$_{NG}$ | SSP5-8.5 (no gradient) | 200 | 500 | 40 |
| LE$_{LMI}$ | SSP1-2.6 | 200 | 500 | 40 - 76 |
| HE$_{LMI}$ | SSP5-8.5 | 200 | 500 | 40 - 76 |
| LE$_{HMI}$ | SSP1-2.6 | 200 | 500 | 40 - 100 |
| HE$_{HMI}$ | SSP5-8.5 | 200 | 500 | 40 - 100 |
| LE$_{MMI}$ | SSP1-2.6 | 200 | 500 | 40 - 135 |
| HE$_{MMI}$ | SSP5-8.5 | 200 | 500 | 40 - 135 |
| LE$_{IC}$ | SSP1-2.6 | 200 - 100 | 500 | 40 |
| HE$_{IC}$ | SSP5-8.5 | 200 - 100 | 500 | 40 |
| LE$_{IC/IM}$ | SSP1-2.6 | 200 - 100 | 500 | 40 - 135 |
| HE$_{IC/IM}$ | SSP5-8.5 | 200 - 100 | 500 | 40 - 135 |

In the relaxation, the values of $\sigma_{\max}$ for both grounded and floating fronts are kept constant at the values specified above. However, calving frequency and magnitude have been observed to vary both seasonally and inter-annually in response to variables such as subglacial discharge, ice mélange or sea ice presence (Todd et al., 2019; Barnett et al., 2022; Slater and Straneo, 2022). Additionally, there are indications of ice tongue weakening in Northern Greenland (Millan et al., 2023), with increased supraglacial melt on Ryder Glacier potentially leading to increased calving of the ice tongue (Holmes et al., 2021).

As such, several experiments emulate a gradual reduction in ice mélange/sea ice strength year-on-year by reducing the value of $\sigma_{\max}$ over the course of a simulation (see Table 1).

### 3.3   Experimental design

In order to isolate the individual impacts of future atmospheric and ocean thermal forcing, different scenarios of SMB and ocean warming were imposed with only one variable changed. Additionally, simulations were run with a combination of different

forcings to understand the combined impact of atmospheric and oceanic forcings. A summary table of all the simulations in shown in Table 1, with more detailed explanations given here.





Two of the simulations, LE (low emissions) and HE (high emissions), impose different SMB scenarios (SSP1-2.6 and SSP5-8.5 respectively) whilst keeping the calving and ocean-driven melt parameters at present-day values. The 'no gradient' (NG) simulations $LE_{NG}$ and $HE_{NG}$ are the same as the former, except that in these simulations - and only these simulations - the SMB-elevation feedback is not included. Three different ocean-driven melt rate increase scenarios are then tested; a low melt increase (LMI), a high melt increase (HMI), and a maximum melt increase (MMI). For the LMI scenario, we assume a linear increase in ocean temperatures relative to present day of $1°C$ by 2100, of $2°C$ by 2200, and of $3°C$ by 2300. Each of these $1°C$ increases corresponds to a 12 m yr$^{-1}$ increase in submarine melt rates, which should be considered a low-end estimate for Ryder Glacier (Wiskandt et al., 2023). For the HMI scenario, we assume the same increase in ocean temperatures, but impose an additional 20 m yr$^{-1}$ of submarine melt per degree of ocean warming - a high-end estimate for Ryder Glacier (Wiskandt et al., 2023). For the MMI scenario, we draw on insights from Wekerle et al. (2024) and impose a nonlinear increase in melt rates for each degree of ocean warming to represent processes such as increases in subglacial discharge with time. Here, we impose an increase of 20 m yr$^{-1}$ for the first degree of warming by 2100, followed by an increase of 25 m yr$^{-1}$ for the second degree of warming towards 2200, and finally an increase of 30 m yr$^{-1}$ for the third degree of warming by 2300. Furthermore, we look into the possibility of increased calving ('IC') in the future by reducing the stress threshold for floating ice linearly with time, both with a constant submarine melt rate (Experiments $LE_{IC}$ and $HE_{IC}$) and in combination with the maximum prescribed increase in submarine melt rates (Experiments $LE_{IC/IM}$ and $HE_{IC/IM}$). In all simulations, the melt rate applied across the entire submarine portion of any grounded fronts is set to equal half of the maximum melt rate below the floating tongue. All of these scenarios are run with SMB forcing from both SSP1-2.6 and SSP5-8.5.

## 4 Results

### 4.1 Sea level rise contribution

The cumulative sea level rise contribution from all low- and high-emission scenarios is shown in Fig. 3. Here, it is seen that the sea-level rise contribution from the low emissions scenarios varies between 0.7 mm and 2.0 mm by 2300. Most of the low emissions scenarios lead to somewhere between 0.8 and 0.9 mm of sea level rise, but simulations $LE_{MMI}$ and $LE_{IC/IM}$ show a higher contribution of nearly 2.0 mm. These simulations follow a similar trajectory up until the late 2200s, after which their mass loss accelerates. The shared feature between these two simulations is that they both include the maximum increase in submarine melt, where a nonlinear relationship between ocean temperatures and basal melt rates is assumed. In both these simulations, the high level of basal melt causes the glacier to retreat into bedrock depression - retreat through which is associated with the rapid increase in mass loss and terminus retreat seen in Fig. 3.

In the high-emissions scenarios, all the simulations with the exception of $HE_{NG}$ show an almost identical trajectory in terms of sea-level rise contribution. By 2300, all of these simulations lead to c. 47 mm of sea-level rise. In contrast, the $HE_{NG}$ simulation leads to 44 mm of sea level rise. All of these sea level contributions are over 40 mm greater than the sea level contribution from any SSP1-2.6 forced simulations. Ryder Glacier currently holds enough ice to contribute 129 mm to sea



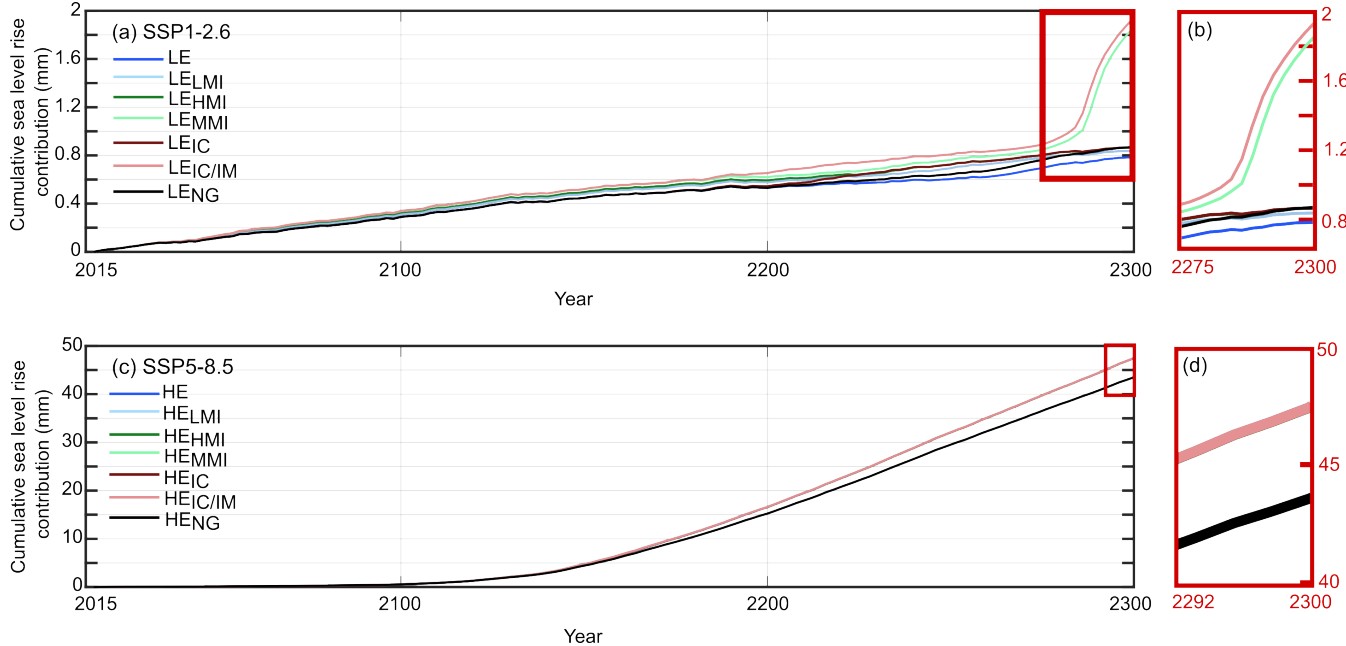

**Figure 3.** Cumulative sea level rise contribution in mm for all low emissions simulations for 2015-2300 is shown in (a), with a zoom-in on 2275-2300 shown in (b). Cumulative sea level rise for all high emissions simulations 2015-2300 is shown in (c), with a zoom-in for 2292 to 2300 shown in (d). In the high emissions simulations, all the simulations except for $HE_{NG}$ ('no gradients') show the same sea level rise contribution curve, meaning that they overlap and it is not possible to discern the different lines on the plot - even when zoomed in. Note the different y axes used for the low and high emissions scenarios.

level rise, meaning that these modelled sea level rise contributions correspond to almost one third of the total sea level rise
245    potential.

## 4.2    Margin retreat and glacier dynamic response

The margin retreat, velocity changes, and thinning magnitude for the LE and $LE_{IC/IM}$ simulations are shown in Fig. 4, and for
HE and $HE_{IC/IM}$ simulations simulations in Fig. 5. These simulations were chosen as they constitute the simulation where only
SMB was changed compared to the spin-up, as well as the simulation in each scenario with the greatest mass loss. Common
250    between all simulations is a trend of retreat and thinning during the period 2015-2300.

In the low emissions scenarios, the ice tongue is always lost by 2300, although it is lost already by 2200 during the two
simulations in which the maximum increase in melt rates in applied ($LE_{MMI}$ and $LE_{IC/IM}$). The magnitude of retreat varies
considerably between low emissions scenarios with the $LE_{IC/IM}$ and $LE_{MMI}$ simulations (where the maximum melt rate increase
is applied) exhibiting nearly 100 km of retreat by 2300, whereas the other scenarios such as the LE simulation (with no melt rate
255    increase) only exhibit around 60 km of retreat. This shows that thinning of the ice tongue from ocean-driven melt is a key driver
of ice-tongue loss in - playing a more important role than increased calving in response to, for example, a reduction in sea ice.



**Figure 4.** Changes in front position, velocity, and ice thickness from LE (a to c) and LE$_{IC/IM}$ (d to f). The thickness change between 2015 and 2300 is shown in panels a and d, where the front positions in 2100 and 2200 are shown. The front position in 2300 corresponds to the limit of the thickness change map. Contours of thickness change are also shown, with thin lines denoting 100m of change and the bold lines denoting 500m of change. Panels b and e show velocity along a flowline with time (note the different y axes scale), and panels c and f show glacier geometry along a flowline with time. The flowline follows the retreating glacier front and is shown as a pale grey line in panels (a) and (d). Bedrock topography, bathymetry, and ice mask data is from BedMachine v5 Morlighem et al. (2017).







**Figure 5.** Changes in front position, velocity, and ice thickness from HE (a to c) and HE$_{IC/IM}$ (d to f). The thickness change between 2015 and 2300 is shown in panels a and d, where the front positions in 2100 and 2200 are shown. The front position in 2300 corresponds to the limit of the thickness change map. Contours of thickness change are also shown, with thin lines denoting 100m of change and the bold lines denoting 500m of change. Panels b and e show velocity along a flowline with time, and panels c and f show glacier geometry along a flowline with time. The flowline follows the retreating glacier front and is shown as a pale grey line in panels (a) and (d). Note that the thickness change scale is different than that used in Fig. 4. Bedrock topography, bathymetry, and ice mask data is from BedMachine v5 Morlighem et al. (2017).

In the simulations showing the most retreat (LE$_{IC/IM}$ and LE$_{MMI}$), the retreat occurs gradually until the retrograde slope located around 80 km along the flowline is reached, after which rapid retreat occurs before re-stabilisation on the prograde slope at 110





km (Fig. 4). By this point, Ryder Glacier's terminus is partly land terminating and partly marine terminating. Although all low
emissions simulations show increasing frontal velocities with time, this rapid retreat along a retrograde slope is associated with
a short-term increase in frontal velocities from circa 1000 m yr⁻¹ to 8000 m yr⁻¹ (Fig. 4). All the low emissions simulations
show a similar pattern of strong thinning of up to 200 m near the glacier terminus, with this extending c. 50 km inland. Further
upstream, areas with both slight thinning and thickening are seen (Fig. 4).

In the high emissions scenarios, the ice tongue is lost in all simulations by 2200, with overall higher levels of retreat and
thinning compared to the low emissions scenarios. Here, ice tongue disintegration is driven by a thinning as a result of the
strongly negative SMB over the ice tongue/lower ablation area. Little variation is seen between the different high emissions
scenarios. In all simulations, including the HE and HE$_{IC/IM}$ simulations shown in Figure 5, the glacier front retreats by over
110 km. Frontal velocities initially increase as the ice tongue is lost, before decreasing after 2200 (Fig. 5be). The entire glacier
domain experiences significant thinning by 2300, with frontal areas showing thinning of around 1 km (Fig. 5ad).

In all simulations, both low and high emissions, the grounding line/grounded terminus position tends to stabilise (or retreat
more slowly) on bedrock highs or prograde slopes (Fig. 4cf and 5cf. In any simulation where the grounding line retreats
upstream of the bedrock sill at around 80 km along the flowline, retreat occurs rapidly (c. 2000 m yr⁻¹) before the glacier
regains some stability on the bedrock high at around 110 km along the flowline.

The retreat of both the grounding line and glacier terminus in LE, LE$_{IC/IM}$, HE, and HE$_{IC/IM}$ are shown in Fig. 6 alongside the
mean frontal velocity from the same simulations. Here, it is clear that under a low-emissions future, the ice tongue may be lost
anytime between 2200 and 2300 depending on ocean forcings - a large temporal range of up to a century. In addition, the mean
frontal velocity tends to increase simultaneously with periods of ice-tongue collapse or rapid terminus retreat, as evidenced by
the spike in frontal velocities up to 900 m yr⁻¹ in the latter half of the 2200s in Fig. 6 panel (a).

This trend is also seen in the high emissions simulations, where velocities are elevated from c. 2050 until c. 2200 during a
period of ice tongue retreat. The data from HE and HE$_{IC/IM}$ show very similar trajectories, but with a offset of c. 10-20 years.

## 4.3 Mass loss partitioning

The partitioning of mass loss between SMB and discharge for several low and high emissions scenarios is shown in Fig. 7, as
well as the mean partitioning of mass loss for different time periods. In a low emissions future, discharge losses are consistently
the driver of overall mass loss - with SMB remaining slightly positive all the way to 2300 as a result of the large accumulation
zone. A large increase in discharge losses around 2280 in LE$_{MMI}$ (Fig. 7b) corresponds to the retreat of the glacier into a deep
basal trough, with these discharge losses starting to stabilise once a bedrock high is reached.

In a high emissions scenario, discharge plays a dominant role in driving mass loss trends up until 2100 after which SMB
becomes more significant and overall mass losses accelerate (Fig. 7). Despite this, discharge losses from the high emissions
simulations remain higher than the discharge losses from the low emissions simulations (Fig. 7). The strongly negative SMB
in the high emissions simulations leads to thinning over the entire model domain (Fig. 5).





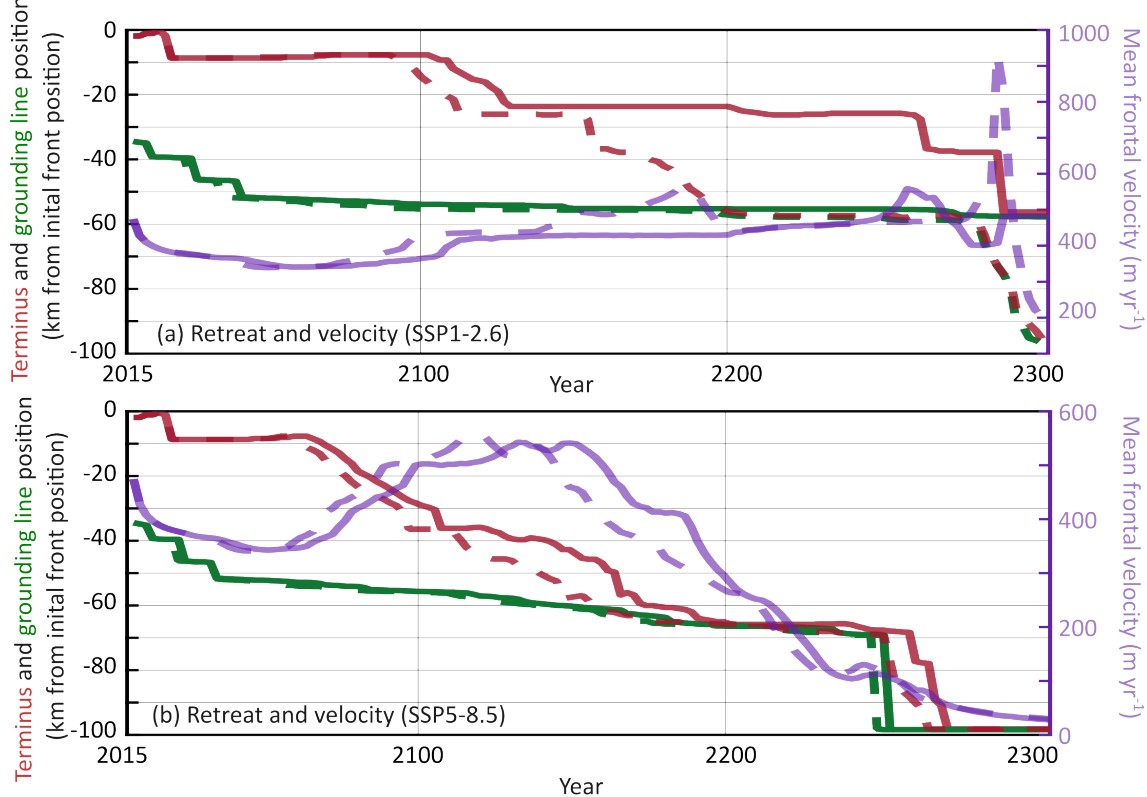

**Figure 6.** Retreat of the glacier front (magenta) and grounding line (green) for low and high emissions scenarios are shown in panels (a) and (b) respectively. The solid lines denote the simulation exhibiting the least mass loss (LE and $HE_{NG}$, and the dashed lines the simulation exhibiting the most mass loss ($LE_{IC/IM}$ and $HE_{IC/IM}$). Where the pink and green lines meet, the glacier no longer has a floating ice tongue. The purple lines denote the mean frontal velocity with time, and uses the y axis on the right hand side of each plot. Note the different y axis scales for the different emissions scenarios.

## 5 Discussions

### 5.1 Influence of emissions scenario

Stark differences can be seen between the simulations using a low emissions scenario and those using a high emissions scenario by 2300 - although all simulations show mass loss dominated by discharge up until 2100. Most importantly, sea-level rise
contributions differ by an order of magnitude. As the suite of low and high emissions simulations were run using the same initialised state, and with the same set of oceanic forcings, the differences between the corresponding low and high emissions simulations can be attributed to the SMB alone. The fact that we see the greatest differences between the simulations with different SMB scenarios, rather than between the simulations forced with different submarine melt rates, suggests that SMB exerts a dominant control on Ryder Glacier's future trajectory. Several Greenland-wide modelling studies have also found that

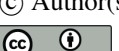



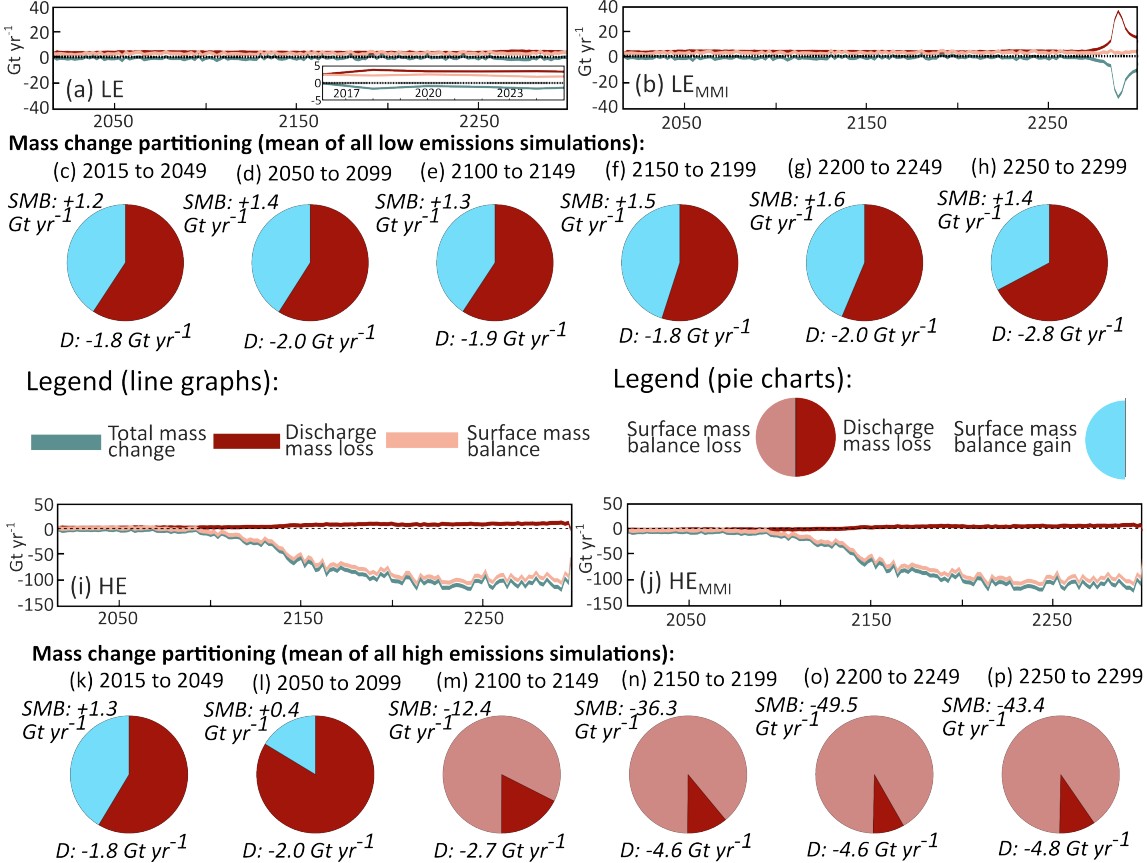

**Figure 7.** Total mass change, SMB mass change, and discharge (denoted 'D') mass change through time for LE (a), LE$_{MMI}$ (b), HE (i), and HE$_{MMI}$ (j). The partitioning of mass change between SMB and discharge is shown for the mean of all low emissions simulations (c to h) and all high emissions simulations (k to p) for the periods 2015-2049, 2050 to 2099, 2100 to 2149, 2150 to 2199, 2200 to 2249, and 2250 to 2300.

SMB is an important driver of future mass loss, but there is significant spread between the magnitude of mass loss between different studies. A study by Aschwanden et al. (2019), which modelled Greenland up until the year 3000, found that the sea-level contribution from Greenland in 2300 was likely to be c. 25 cm under RCP 2.6 (comparable to SSP1-2.6) and c. 174 cm under RCP 8.5 (comparable to SSP5-8.5). These sea-level rise contributions correspond to 3.4% and 24% of Greenland's sea-level rise potential respectively. In the simulations presented here, the sea-level rise contribution from Ryder Glacier is 0.08 - 0.2 cm under SSP1-2.6 and 4.4 - 4.7 cm under SSP5-8.5. The total potential sea-level rise contribution from Ryder Glacier is 12.9 cm, meaning that these values correspond to 0.6 - 1.5% (SSP1-2.6) or 34 - 37% (SSP5-8.5) of Ryder Glacier's sea-level rise potential. Therefore, our simulations show a smaller percentage-wise sea-level rise contribution under SSP1-2.6 compared to the Greenland wide study, but a larger percentage-wise sea level rise contribution under SSP5-8.5. However, our sea-level rise contribution under SSP5-8.5 is smaller than that found by another Greenland-wide modelling study by





Beckmann and Winkelmann (2023). Here, the authors found that Greenland is likely to contribute between 3 and 3.5 m to global sea-level rise by 2300 - which is 41 - 49% of the total potential sea-level rise contribution from Greenland (Beckmann and Winkelmann, 2023). It should be noted that the study by Beckmann and Winkelmann (2023) focuses on extreme events and so likely constitutes an end member scenario. However, the comparison between our results and the two Greenland-wide modelling studies shows that the results presented in this paper for Ryder Glacier fall within the range of results from previous

studies. The differences between our results and the aforementioned papers is also partly due to the fact that the development of Ryder Glacier as an individual outlet glacier is not expected to follow the same trajectory as the Greenland ice sheet as a whole. This may be because the higher mesh resolution employed here on the smaller domain allows for site specific characteristics such as bed topography to be well resolved, allowing for behaviours related to marine ice sheet instability to be fully captured by the model. As such, running high resolution studies that focus on individual glaciers or regions is vital for validating the

results from broader-scale studies that must, for computational reasons, employ a coarser grid resolution.

The simulations forced with a SSP5-8.5 future showed less sensitivity to variations in ocean forcing than the simulations forced with an SSP1-2.6 future. In Fig. 3, all the high emissions simulations follow the same indistinguishable trajectory with the exception of HE$_{NG}$, where no SMB-elevation feedback was incorporated. This relationship between glacier behaviour and ocean melt under different emissions scenarios is something not considered in other studies, where the Greenland-wide focus

meant ocean-induced melt rates were fixed in time (Beckmann and Winkelmann, 2023) rather than increasing in line with results from site-specific modelling and observational studies (Wilson et al., 2017; Wiskandt et al., 2023). The results of HE$_{NG}$ are very similar to the other high emissions scenarios until around c. 2150, after which they start to diverge, corresponding to the onset of extensive thinning at Ryder Glacier. The SMB-elevation feedback has been previously identified as a key driver of runaway melt and retreat of the Greenland ice sheet (Bochow et al., 2023), something that we also find here. Specifically,

our results show that the exclusion of this feedback has a greater impact on the modelled sea level rise contribution results than the chosen basal melt rate scenario, at least for the high emissions simulations. The SMB-elevation feedback is likely particularly important for glaciers in the Northern sector of the Greenland ice sheet as the gently sloping margins in the area result in some of the fastest rates of upward migration of the ablation zone and runoff line (the upper extent of where runoff exceeds 100 mm w.e yr$^{-1}$) across all of Greenland (Noël et al., 2022). In combination with this, the interior of the Northern

sector has a generally lower elevation than the Northeastern and Northwestern sectors, a characteristic which lends itself to accelerated mass loss (Noël et al., 2022). Incorporating this SMB-elevation feedback into models has led to increases in the expected mass loss from the Northern sector (Muntjewerf et al., 2020), something also seen in this study when comparing the HE (SMB-elevation feedback included) and HE$_{NG}$ (no SMB-elevation feedback) simulations. In the low emissions simulations, the simulation without the SMB-elevation feedback (LE$_{NG}$) fits well within the range of the other low emissions simulations -

likely reflecting the minor effect of the SMB-elevation feedback in a situation where thinning of the ice sheet surface is much more limited.



## 5.2   Drivers of mass loss

In the low emissions simulation, there is a clear increase in frontal velocities associated with front retreat and ice-tongue loss. For example, in Fig. 4b, the frontal velocities increase from around 750 m yr$^{-1}$ to around 1500 m yr$^{-1}$ within the last 50 years or so of the simulation; corresponding to the complete loss of the ice tongue as shown in Fig. 4c. However, this effect is not only seen when the ice tongue is lost, but also during periods of rapid retreat of the grounded glacier front. This is exemplified in Fig. 4e/f where the rapid retreat of Ryder Glacier across a bathymetric trough in the late 2200s is associated with frontal velocities that reach over 8000 m yr$^{-1}$. This is an over fourfold increase in frontal velocity (see Fig. 6), and is associated with a thinning of the glacier frontal region on the order of several hundred metres over a period of c.15 years. This period of acceleration and thinning begins where the then-grounded glacier front retreats onto a retrograde slope and into a bathymetric depression, thus being an example of marine ice-sheet instability (Schoof, 2007; Schoof et al., 2017) This period of rapid retreat, acceleration, and thinning is only seen in the LE$_{MMI}$ and LE$_{IC/IM}$ simulations, where the increased ocean-driven melting is sufficient to retreat the glacier onto the retrograde slope. By 2300, the glacier has somewhat stabilised on the prograde slope at the landward side of the bathymetric depression, with subdued retreat rates and a decrease in frontal velocities down to c. 4000 m yr$^{-1}$. However, by this point, the discharge losses from these simulations have caused a spike in mass loss and sea-level rise contribution, as can be seen in Fig. 3. Although Ryder Glacier is still grounded below sea level, the basal topography at this point is shallow and, in parts, even above sea level. Further retreat would lead to a greater proportion of Ryder Glacier becoming land terminating, meaning that it is unlikely to undergo any further periods of rapid ocean-driven mass loss.

Although all the high emissions simulations also show Ryder Glacier retreating through this bathymetric trough, there is no clear dynamic response in these simulations. Instead, both the simulations shown in Fig. 6 show acceleration occurring around 2200, coincident with the loss of Ryder Glacier's ice tongue. When the glacier then retreats through the bathymetric trough, the glacier has already thinned so much due to a negative SMB (i.e., surface melt) that mass loss is dominated by down-wasting near the front rather than ocean-driven retreat. This is combined with the fact that Ryder Glacier has retreated to shallower ground by this point and is, in parts, land terminating. Only the central part of the terminus remains marine terminating, with the entire glacier projected to become grounded above sea level with an additional c. 9 km of retreat. As a result, ocean driven forcing is not as important for the overall mass balance in 2300 as it is in the present day. However, there is still evidence of bathymetric control up until c. 2100; the retreat of the grounding line from its present-day position up the prograde slope is interrupted by short stillstands on small bathymetric ridges. This trend of an acceleration towards 2200 followed by a deceleration is consistent with patterns found by Beckmann and Winkelmann (2023) for the Northern sector from a Greenland-wide modelling study. In their study, extensive thinning and the associated decrease in driving stress was suggested as the cause of slower velocities after 2200 - something that we also propose as the reason for the reduced velocities after c. 2200 here.

Considered together, both the low- and high-emission simulations provide evidence for a dynamic response to ice-tongue loss at Ryder Glacier. This suggests that the ice tongue is currently exerting a buttressing force on the grounded areas of the glacier, and suggests that mass loss and sea-level rise contribution from Ryder is likely to increase in conjunction with a potential future





ice-tongue disintegration. This at first appears to be in contrast to observations from the neighbouring Petermann Glacier, where large calving events in 2010 and 2012 have been associated with a limited dynamic response (Münchow et al., 2014; Nick et al., 2012). However, modelling work at Petermann Glacier shows that potential future calving events closer to the grounding line are likely to cause large increases in velocity and discharge (Hill et al., 2018a; Åkesson et al., 2022). At Ryder Glacier, the

modelling experiments presented here all exhibit total ice-tongue collapse with these events therefore being, by definition, in close proximity to the grounding line. As such, the associated accelerations and increase in discharge losses are similar to those projected to occur under similar circumstances at Petermann Glacier. This result suggesting a dynamic response after ice tongue collapse at Ryder glacier may therefore provide insight into how other Greenlandic glaciers with ice tongues may behave in the future, especially important given that previous studies have found that model uncertainties are higher for glaciers

with ice tongues in Greenland-wide setups (Choi et al., 2021).

    We find clearly contrasting drivers of mass loss across the suite of low emissions simulations compared to the suite of high emissions simulations. As shown in Fig. 7, Ryder Glacier experiences an overall positive SMB during the entirety of the modelled period, although the frontal region of the glacier experiences a net negative SMB. This may be in contrast to trends over the entire Northern sector, where SMB has been net negative since around the year 2000, but when considering

Ryder Glacier by itself there has been a net positive SMB of 2.1 Gt yr$^{-1}$ in the period 2000-2017 (Mouginot et al., 2019). In the simulations presented here, net SMB is lower than the 2000-2017 mean but does not change significantly throughout the study period with discharge losses driving overall mass loss. The magnitude of these discharge losses consistently outweighs the positive surface mass balance leading to an overall mass loss of around 0.65 Gt yr$^{-1}$ averaged across the entire study period. This is lower than the mean mass balance of -0.96 Gt yr$^{-1}$ for the period 2000-2017 but significantly more negative

than the mean mass balance of +0.33 Gt yr$^{-1}$ observed during the period 1972-1990. The lack of an increased mass loss under a low emissions scenario likely reflects the fact that the accumulation zone of Ryder Glacier is large and retains a net positive SMB during the entire study period. As the ice tongue is lost and Ryder Glacier's front retreats by several tens of kilometres, large swathes of the current ablation zone are removed. At the same time, the shallowing basal topography inland of Ryder Glacier's present day grounding line - albeit it with some basal troughs - means that the impact of ocean forcing and discharge

driven mass loss is constrained. Together, these features (which are specific to the bathymetry of Ryder Glacier and to the low emissions scenario), act to modulate future mass loss. This geometric control on Ryder Glacier's future behaviour highlights the need for high-resolution studies that can resolve topographic features under individual outlet glaciers, providing results which can inform and validate Greenland-wide studies.

    The pattern of SMB and discharge trends is significantly different for the high emissions simulations, where the SMB is

positive until 2099, after which it decreases significantly to become the dominant driver of mass loss from 2100 onwards. Discharge losses are, like in the low emissions simulations, a key driver of mass loss until 2099, after which they are dwarfed by SMB-driven losses. By 2300, SMB losses are an order of magnitude higher than discharge driven losses and little variation is seen between simulations. Under SSP5-8.5, temperatures in the Arctic are projected to increase drastically by 2100, becoming c. 10$^{\rm o}$C warmer than during the period 1995-2014 (Lee et al., 2021). This is significantly higher than the expected global

temperature anomaly of 4$^{\rm o}$C, as a result of ongoing Arctic amplification (Rantanen et al., 2022). This is additionally higher





than projected temperature changes under SSP1-2.6, where the Arctic is expected to experience temperatures 2.4°C warmer than mean 1995-2014 values, with the global mean temperature anomaly only expected to be 0.7°C (Lee et al., 2021). The gap between these scenarios continues to grow past 2100 and, in the case of SSP5-8.5, the lowering of the ice sheet surface further exacerbates the local SMB anomaly. As such, Ryder Glacier is likely to exhibit extensive thinning across its entire drainage

basin under an SSP5-8.5 future, with discharge losses playing a relatively insignificant role despite being considerably higher than in the present.

These results suggest that the future emissions path, and associated atmospheric warming, exerts a dominant control on overall mass loss, with the largest variations seen between the low and high emissions simulations rather than within them. However, discharge will play a key role in driving mass loss at Ryder until 2100 regardless of emissions scenario. This result

corroborate well with that of Choi et al. (2021), who found that ice dynamics would likely be a key driver of mass loss from Greenland during the next century, but with SMB playing an increasingly important role as time goes on. This study adds to these conclusions by continuing simulations past 2100, showing that SMB-driven loss rapidly surpasses discharge loss post 2100 in a high emissions future. Additionally, bathymetry plays a significant role in determining shorter term variability, with periods of acceleration and elevated mass loss being associated with retrograde slopes and deeper basins.

## 5.3   Comparisons with other glacier-fjord systems

The finding that Ryder Glacier's ice tongue collapse leads to a dynamic response at first appears to be at odds with observations from the neighbouring Petermann Glacier, where large calving events in 2010 and 2012 have been associated with a limited dynamic response (Münchow et al., 2014; Nick et al., 2012). However, modelling work at Petermann Glacier shows that potential future calving events closer to the grounding line are likely to cause large increases in velocity and discharge (Hill

et al., 2018a; Åkesson et al., 2022). At Ryder Glacier, the modelling experiments presented here all exhibit total ice-tongue collapse with these events therefore being, by definition, in close proximity to the grounding line. As such, the associated accelerations and increase in discharge losses are similar to those projected to occur under similar circumstances at Petermann Glacier. This is in contrast to the limited dynamic response response observed after the 2002–2003 collapse of C.H. Ostenfeld Glacier's ice tongue (Hill et al., 2018a). This likely links back to ice tongue/ fjord geometry, where the buttressing afforded by

an ice tongue is likely lower in glacier-fjord systems where there is little contact between the lateral margins of the ice tongue and the fjord walls - as is seen at C.H. Ostenfeld Glacier prior to collapse. These findings are relevant to Greenland's largest ice tongue, 79N (Nioghalvfjerdsfjorden) Glacier, which has experienced a similarly limited volume change as Ryder Glacier in recent decades, despite evidence of warm Atlantic Intermediate Water near its grounding line (Millan et al., 2023; Bentley et al., 2023). 79N Glacier's ice tongue is attached to the adjacent fjord walls, with the ice tongue narrowing c. 20 km inland of

its present-day terminus position. Our model findings for Ryder Glacier would suggest that the fjord geometry at 79N Glacier may be helping to keep the terminus position relatively stable despite being subject to the highest basal melt rates out of all the Greenlandic glaciers with ice tongues (Wekerle et al., 2024). However, the strong traction between ice tongue and fjord walls at 79N Glacier also suggests that the ice tongue plays an important role in terms of buttressing, with evidence of acceleration in response to thinning having already been documented (Khan et al., 2022).





These findings also have implications for glaciers without ice tongues, and highlight the important role of fjord geometry in moderating glacier response to external forcings (Catania et al., 2018). The impact of fjord geometry on glacier stability has been previously studied, although often with the use of simplified synthetic geometries (e.g. Åkesson et al., 2018; Frank et al., 2022). The differing behaviour of Ryder, Petermann, and C.H. Ostenfeld glaciers showcased here highlights how these topographical controls can lead to highly heterogenous behaviour in real-world neighbouring systems. The rapid retreat of

Ryder Glacier in the low emissions scenarios once a retrograde bed slope is reached illustrates that seemingly stable glaciers can suddenly exhibit rapid retreat, having been in a state of disequilibrium with the ambient climate (Robel et al., 2022). Understanding these processes is vital for accurately forecasting the timing of glacier retreat and associated sea level rise (Robel et al., 2022). This suggests that once a threshold is reached - which may occur asynchronously with climate forcing due to geometry-imposed stability or as a result of sustained change over several years - drastic increases in velocity and discharge

losses may occur. These losses are inherently difficult to predict, not least considering the uncertain shape of the seafloor and under-ice topography. This underscores that continued, targeted efforts to map bathymetry and subglacial topography for the polar ice sheets are crucial. Such observational efforts should be accompanied by models with a fine enough resolution to capture these topographic peaks and troughs, which we show here exert a very strong control on ice-tongue and grounding-line stability. The results presented here add to our understanding of these processes by showing these rapid, non-linear increases in

mass loss to be of particular importance also in a low-emissions future, with uncertainty in the timing of rapid retreat dominated by the choice of basal melt rate scenario. As sea level rise in the coming decades (up to 2100) need to be planned for policy makers in the near future, it is imperative that the processes driving mass loss in this time frame are well understood and constrained. As such, improving the representation of ocean driven melt rates as well as running numerical simulations at a spatial resolution that can resolve fjord and bedrock geometry is vital for helping to reduce uncertainties in future projections.

## 5.4   Model Limitations

Whilst providing useful projections of the future, all models have limitations. The Blatter-Pattyn/HO approximation of ice flow used in this study neglects bridging stresses, which is valid over the majority of a glacier or ice sheet. Despite this, errors may arise in areas where large velocity gradients are found over short distances - such as at grounding lines. Various studies have found that Blatter-Pattyn stress regimes can lead to either an overestimation of velocities/grounding line discharge (Rückamp

et al., 2022) or lower velocities and subdued grounding line acceleration (Yu et al., 2018). However, a comparison of model results at Thwaites Glacier (which has a floating extension) with the same 300m maximum resolution as in our study found little difference between the overall future trajectories modelled with Full-Stokes and HO stress balance approximations (Yu et al., 2018). This suggests that the errors arising from the use of a HO model do not negate conclusions surrounding long term changes and their driving processes which, when combined with the fact that the computational costs of using Full Stokes

would limit the number/length of simulations we could run, guided our choice of a HO model. The choice of sliding law and calving law used in this study was guided by previous work, which focused on assessing the fidelity of various sliding and calving laws in models of Greenlandic tidewater glaciers and ice tongues (Choi et al., 2018; Åkesson et al., 2021). However, there remains uncertainty surrounding how the characteristics of any given glacier may change moving into the future. For





example, recently identified supraglacial streams on Ryder Glacier's ice tongue have been linked to increased calving that may mean the calibrated value of $\sigma_{max}$ used as the initial value in this study is too high (Holmes et al., 2021). Future work focusing on how best to incorporate changes in calving rates due to supraglacial melt and the loss of buttressing from reduced sea ice/ ice mélange would be beneficial. Furthermore, the incorporation of basal/frontal melt rates from ocean models would allow for variation in melt rates across the width of the ice tongue/glacier front and for temporal evolution on seasonal and interannual time scales. Incorporation of more realistic basal melt would likely lead to higher rates of mass loss (Christian et al., 2020), which suggests that our computed mass loss under the low-emissions scenarios should be considered conservative estimates. In addition, the shallow bathymetric sill which sits in front of Ryder Glacier's grounding line is documented to block intrusions of warm Atlantic Water (Jakobsson et al., 2020). Importantly, this reduces the basal melt as well as its sensitivity to changes of the Atlantic Water temperature (Nilsson et al., 2023); a potentially stabilising effect that has for simplicity not been included in the present basal-melt scenarios. These calving and basal melt uncertainties are of particular importance in light of the fact that recent reports suggest Greenland's northern ice tongues may be undergoing rapid weakening (Millan et al., 2023).

## 6 Conclusions

We have conducted targeted, high-resolution simulations of the future fate of Ryder Glacier, which has been relatively stable in recent years compared to neighbouring glaciers. These simulations are forced with future projections of ocean forcing, surface mass balance, and calving. Our model results suggest that up until the year 2100, discharge losses are likely to play a dominant role in driving mass loss at Ryder Glacier regardless of emissions scenario. Under an low-emissions future (SSP1-2.6), discharge will continue to be the driver of mass loss until 2300. However, a strongly negative SMB under high-emission future (SSP5-8.5 ) will cause SMB to take over as the primary driver of mass loss, and lead to a sea-level rise contribution of over 45 mm. The bedrock topography and shape of Sherard Osborn fjord play a vital role in both the present day and future scenarios. These topographic factors are particularly important under a low-emissions future where bedrock depressions are associated with periods of significantly increased mass loss. This suggests that high-resolution studies at the single glacier or regional scale are important for constraining uncertainties in the timing and magnitude of future contributions to sea-level rise from the Greenland Ice Sheet. Ryder Glacier is likely to lose its ice tongue under all simulated scenarios by 2300, but with ice-tongue loss occurring up c. 80-100 years earlier under a high emissions scenario, or where basal melt rates are increased more rapidly. Overall, the results suggest that retreat and ice-tongue breakup will occur even under a low-emission future climate. However, the latter future pathway will likely lead to much lower sea-level rise contribution, and a greatly reduced negative societal impact, when compared to a high-emission future climate.

*Code and data availability.* All the code necessary for running simulations using ISSM is freely available at https://issm.jpl.nasa.gov/download/. The necessary surface mass balance forcing data for running the simulations is available at https://zenodo.org/records/7100706, and we are grateful to Brice Noël and collaborators for providing this.



*Author contributions.* The research questions were designed by FAH and MJ with help from all authors. The model was set-up by FAH, JB, HÅ, and MM with JN, NK, and MJ also contributing to the experimental design. The results were discussed by all authors. The first draft of the manuscript was written by FAH, with subsequent input from all authors.

*Competing interests.* The authors declare no competing interests

*Acknowledgements.* The running of all simulations was enabled by resources provided by the National Academic Infrastructure for Super-
computing in Sweden (NAISS) at the National Supercomputer Centre at Linköping University, Sweden, partially funded by the Swedish Research Council through grant agreement no. 2022-06725. FAH was funded by Formas Grant no. 2021-01590 awarded to MJ. HÅ was funded by the project JOSTICE from the Norwegian Research Council (grant #302458), and by ERC-2022-ADG grant agreement No 01096057 GLACMASS from the European Research Council.



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
