# Peer review of "Sea-level rise contribution from Ryder Glacier in Northern Greenland varies by an order of magnitude by 2300 depending on future emissions"

_EGUsphere, 2024_

## Referee Comment (RC1)

**Review of Holmes et al. 'Sea-level rise contribution from Ryder Glacier in Northern Greenland varies by an order of magnitude by 2300 depending on future emissions'**

**General comments**

The authors present catchment-scale numerical ice sheet modelling simulations designed to make projections of potential future sea level rise contribution from Ryder Glacier. Unlike surrounding glaciers in northern Greenland, Ryder Glacier's ice tongue remains largely intact and has not shown retreat and/or collapse in recent decades, raising the quesion as to why that might be. The authors run a suite of experiments exploring the future of Ryder glacier under two SSP forcing scenarios alongide the sensitivity to calving and basal melt rates. They find the scenario dependence has a greater impact on sea level rise contribution by 2300 than the exact prescription of basal melt rates or an increase in calving rate through time. All of their simulations find the ice tongue to collapse in the future.

Targeted numerical modelling experiments have not been conducted - to my knowledge - on this glacier before, and so the research is timely and important. I think the modelling framework is good and the results are well presented. I do however have some comments to improve the manuscript, including more details on the initialisation and sensitivity to the choice of calving law. These more major comments are detailed in the numbered list below and specific line comments are in the following Section.

1. **Abstract:** the entire abstract, except for the final sentence is background/method/aim. This needs revising. Normally abstracts would be weighted towards a couple of sentences of background and motivation, the same again for the aim and methods, and then most on the new findings and implications. It does not currently include your main findings, i.e. the range of potential sea level contribution by 2300 nor the scenario dependence stated in the title.

2. **Introduction:** The aims at the end of the Introduction do not reflect the title of the paper, nor the work of the paper. These are a bit confusing to the reader, because it suggests that you study all three glaciers and provide an answer for why these glaciers are behaving differently. While you can gain insights into that from the work you do here, you are only modelling Ryder Glacier. The aims at the end of Section 2 are a much better reflection of the aims/objectives that are actually addressed in this paper. See the following point, but make sure that the reader is easily able to find the main question/aim of the study in the final paragraph or so of the Introduction.

3. **Section 2:** I think this section is unnecessary, and detracts from the main focus of the paper. It leads the reader to think that you will go on to model all three glaciers to make a thorough comparison. Again, as mentioned above, this Section additionally causes confusion because you have one set of aims at the Introduction and a second set at the end of this section. My recommendation is, you move some of the material in Section 2 into the Introduction, e.g. lines 95-115 to highlight the importance of studying Ryder Glacier, given how differently it has been behaving to the neighbouring glaciers. You could perhaps make some initial statements as to why that might be e.g. SMB and fjord conditions. Then the rest of the comparison should come in the Discussion, in Section 5.3. There is no need for the level of detail in this section, and a lot of this could be summarised in a few sentences with signposting to other studies.

4. **Model relaxation and calving:** I would have liked to see a bit more on how your model performs during the 50-year relaxation period, I assume that the trend must be very similar to observations to justify not having a control run from which to subtract from the perturbed simulations (as is often done, e.g. in the ISMIP experiments, to account for model drift).

   As part of this relaxation I think more detail on the choice of calving law and what values of $\sigma_{max}$ were testing to arrive at 200 kPa should be included in the Methods section, preferably with some sensitivity tests and additional figure. Is this calving law really able to replicate the observed

changes in front position of Ryder Glacier over the last 50 years, which is characterised by a cycle of slow advance and then a large calving event before re-advance? Do you expect that calving style to continue into the future and does this calving law reflect that? I think all of these warrant some explanation if not in the methods, but as limitations in the Discussion.

In addition, while you account for the impact of increased calving (IC) in some of your experiments, you do not explore the low-end member of no calving. I think there is some scope for including a control run, or some exploration of the sensitivity of your results to this choice of calving law and chosen value of $\sigma_{\max}$.

**Specific comments**

**Line 1:** how many glaciers with floating ice tongues still exist outside of northern Greenland? if none, change 'some of' to 'all'.

**Line 5:** suggest changing 'increased mass loss from discharge' to just 'increased ice discharge'

**Line 33:** State some references of studies that have presented these 'observational records'.

**Line 40:** Throughout I would clearly state what you mean by 'stable' i.e. 'mass balance is close to zero' or 'limited change in front position'

**Figure 1:** See comment above about Section 2. I think this study figure should focus on Ryder Glacier and there is no need to necessarily include the others. Also the inset currently covers half of Ryder Glacier's catchment. Possibly show the model domain/mesh in this figure and the flowline used in other figures. I think the colour bar should be labelled speed rather than velocity.

**Line 116:** Heading 'Numerical model' doesn't really make sense given that this section includes a large part on input data. Perhaps consider changing to Methods and then 3.2 could be 'Model-set up' or similar.

**Line 120:** Does the domain allow for advance during the relaxation?

**Line 136:** State the resolution of the velocity dataset.

**Line 138:** I am not totally convinced by this justification for using Budd friction law, is there any reason to suggest that because it works best for Petermann it would for Ryder? I think if not testing different sliding laws in this study, the potential impact of this choice needs to be in the Discussion, especially given that Akesson et al. 2021 show this can have a large impact on future projections at Petermann.

**Line 145:** Is it common to have a spatially uniform viscosity field over the grounded ice? Perhaps mention some other studies that have used this approach. It may also warrant a space in the limitations section of the Discussion.

**Lines 146-156:** I think these lines would be better placed in a section 'Relaxation' because it is currently confusing to discuss this before introducing the calving parameterisation Eq. 2 and introducing the SMB forcing in Section 3.2.1.

**Figure 2:** If possible make panel d bigger so it's possible to see some of the detail in the change through time, although I appreciate the geometry is not changing much during this relaxation.

**Line 166:** More details on these SSP forcing are needed. Which CMIP model was used? are these an ensemble mean? Also do you apply these as anomalies ontop of the mean SMB field used during the relaxation?

**Line 176:** What does a 'high level of subglacial discharge' equate to?

**Line 199:** I'm not sure what you mean by 'Ryder Glacier's grounded calving front during the relaxation', surely the calving front is floating during the relaxation? make this clearer to the reader.

**Line 206:** How was the value of $\sigma_{\max}$ varied throughout the simulation? linearly reduced each year?

**Line 228:** when you say 'grounded fronts' do you mean when the ice fronts become grounded during the experiment when the ice tongue is lost? I think a better explanation of what you mean by grounded fronts throughout would be useful.

**Figures 4 and 5:** units for thickness change.

**Figure 6:** Where is this frontal velocitiy taken from? Is this the most useful measure? I would have instead reccomended to show the change in velocity inland of the terminus, to show how changes at the front (thinning/calving) impact inland grounded ice flow.

**Figure 7:** This Figure needs a bit of a rethink, the lines in panels a, b, i, j are impossible to see any trend in and are too small. I think the sign of the red line in b) is wrong if you are expressing discharge as a loss.

**Section 4.3:** was the integrated SMB calculated over the grounded area only? and how was discharge calculated? across the grounding line? or a defined flux gate?

**Line 286:** I find 'discharge losses' to be a bit awkward to read, I suggest changing it throughout to 'increases or decreases in ice discharge'.

**Line 298:** Surely the fact that the ice tongue was lost quite quickly in the high-end simulations is one of the reasons for the submarine melt rates not having such an impact in these simulations. Perhaps worth mentioning.

**Lines 302-315:** All of this could be shortened and summarised in a few sentences. Also if making a comparison between your study and other Greenland wide results I suggest including the ISMIP6 results.

**Line 321:** See comment above, surely the reason there is less sensitivity to ocean forcing is because at some point there is no ice tongue left? Worth mentioning this.

**Line 337:** The impact of the melt-elevation feedback may not be straightforward. Consider adding a reference to Delhasse et al., 2024 - TC who showed in coupled simulations the positive melt-elevation feedback may be mitigated.

**Line 380:** 'total ice-tongue collapse' to me suggests instantaneous collapse, i.e. the timeframe of C.H. Ostenfeld ice tongue collapse, whereas as far as I understand in your experiments the removal of the ice tongue is gradual due to the calving law? in which case I'm not sure have replicated an entire/immediate tongue collapse and it would be worth discussing this.

**Line 392:** see comment above, I suggest using 'increased ice discharge' instead of 'discharge losses' throughout.

**Lines 426-432:** These lines are an exact duplicate of Lines 377-382. I suggest a careful proof read of the entire Discussion making sure there is no duplication and that the information presented is as concise as possible.

**Line 455-460:** This section on topography is important, but I do wonder about mentioning that loss of the ice tongue doesn't appear to initiate a runaway retreat of the grounding line further inland.

**Section 5.4:** I think the limitations on choosing a single sliding law and calving law need to be discussed in more depth here.

**Line 505:** This sentence would benefit from rephrasing. State the pathway instead of using 'latter'. 'greatly reduced negative societal impact' reads awkwardly as well.

---

## Author Response (AR1)

**We are grateful to both reviewers for the comprehensive and constructive comments on the manuscript. Below, we have detailed our point-by-point responses (in blue) to all of the comments. We have also uploaded a revised version of the manuscript, as well as a manuscript file with tracked changes.**

**Yours sincerely,**

**Felicity Holmes (on behalf of all co-authors)**

**Reviewer 1:**

*General comments*
The authors present catchment-scale numerical ice sheet modelling simulations designed to make projections of potential future sea level rise contribution from Ryder Glacier. Unlike surrounding glaciers in northern Greenland, Ryder Glacier's ice tongue remains largely intact and has not shown retreat and/or collapse in recent decades, raising the quesion as to why that might be. The authors run a suite of experiments exploring the future of Ryder glacier under two SSP forcing scenarios alongide the sensitivity to calving and basal melt rates. They find the scenario dependence has a greater impact on sea level rise contribution by 2300 than the exact prescription of basal melt rates or an increase in calving rate through time. All of their simulations find the ice tongue to collapse in the future.

Targeted numerical modelling experiments have not been conducted - to my knowledge - on this glacier before, and so the research is timely and important. I think the modelling framework is good and the results are well presented. I do however have some comments to improve the manuscript, including more details on the initialisation and sensitivity to the choice of calving law. These more major comments are detailed in the numbered list below and specific line comments are in the following Section.

Thank you for your helpful review on our manuscript. We appreciate your time and have detailed our responses to your comments below.

1. Abstract: the entire abstract, except for the final sentence is background/method/aim. This needs revising. Normally abstracts would be weighted towards a couple of sentences of background and motivation, the same again for the aim and methods, and then most on the new findings and implications. It does not currently include your main findings, i.e. the range of potential sea level contribution by 2300 nor the scenario dependence stated in the title.

The abstract has been changed to remove some of the background/motivation, and to add in more information on the key results. The new abstract is shown below:

*The Northern sector of the Greenland ice sheet contains some of the ice sheet's last remaining glaciers with floating ice tongues. One of these glaciers is Ryder Glacier, which has been relatively stable in recent decades in contrast to the neighbouring Petermann and C.H. Ostenfeld Glaciers. Understanding Ryder Glacier's future behaviour is important as ice-tongue loss could lead to acceleration and increased ice discharge. Meanwhile, it is unclear whether Greenland-wide modelling attempts are able to accurately resolve the influence of fjord/bedrock topography and small-scale variations in ice dynamics for a glacier like Ryder. To fill these gaps, we here conduct targeted high-resolution modelling to*

*Ryder Glacier until the year 2300. We find that mass loss is dominated by discharge under a low emissions scenario all the way to 2300, leading to a sea level contribution of between 0.8 to 2 mm depending on the degree of ocean warming. Discharge also plays a key role under a high emissions scenario up until 2100, after which a strongly negative surface mass balance becomes the dominant driver of mass loss. This negative surface mass balance leads to much higher sea level rise contribution by 2300 of between 44 and 52 mm, with little sensitivity to the range of ocean warming scenarios used in this study.*

2. Introduction: The aims at the end of the Introduction do not reflect the title of the paper, the work of the paper. These are a bit confusing to the reader, because it suggests that you study all three glaciers and provide an answer for why these glaciers are behaving differently. While you can gain insights into that from the work you do here, you are only modelling Ryder Glacier. The aims at the end of Section 2 are a much better reflection of the aims/objectives that are actually addressed in this paper. See the following point, but make sure that the reader is easily able to find the main question/aim of the study in the final paragraph or so of the Introduction.

Thanks for this comment and related suggestions – similar issues were also raised by the other Reviewer. We have removed section 2 (Petermann, Ryder and C.H. Ostenfeld glaciers), replaced the study area figure with one focusing on Ryder glacier. Some material from this section has been incorporated into the introduction and discussion section 'Comparison with other Glacier-Fjord systems'.

3. Section 2: I think this section is unnecessary, and detracts from the main focus of the paper. It leads the reader to think that you will go on to model all three glaciers to make a thorough comparison. Again, as mentioned above, this Section additionally causes confusion because you have one set of aims at the Introduction and a second set at the end of this section. My recom- mendation is, you move some of the material in Section 2 into the Introduction, e.g. lines 95-115 to highlight the importance of studying Ryder Glacier, given how differently it has been behaving to the neighbouring glaciers. You could perhaps make some initial statements as to why that might be e.g. SMB and fjord conditions. Then the rest of the comparison should come in the Discussion, in Section 5.3. There is no need for the level of detail in this section, and a lot of this could be summarised in a few sentences with signposting to other studies.

Please see response to comment number 2.

4. Model relaxation and calving: I would have liked to see a bit more on how your model performs during the 50-year relaxation period, I assume that the trend must be very similar to observations to justify not having a control run from which to subtract from the perturbed simulations (as is often done, e.g. in the ISMIP experiments, to account for model drift). As part of this relaxation I think more detail on the choice of calving law and what values of σmax were testing to arrive at 200 kPa should be included in the Methods section, preferably with some sensitivity tests and additional figure. Is this calving law really able to replicate the observed changes in front position of Ryder Glacier over the last 50 years, which is characterised by a cycle of slow advance and then a large calving event before re-advance? Do you expect that calving style to continue into the future and does this calving law reflect that? I think all of these warrant some explanation if not in the methods, but as limitations in the Discussion. In addition, while you account for the impact of increased calving (IC) in

some of your experiments, you do not explore the low-end member of no calving. I think there is some scope for including a control run, or some exploration of the sensitivity of your results to this choice of calving law and chosen value of σmax.

The entire methods/model description has been re-written, including a new subsection focusing solely on the model relaxation. Here, we have made the aims of our relaxation clearer, provided more information on how the calving calibration was conducted, and compared our relaxation results to observations in a more explicit manner. As part of this reshuffle, the calving law is introduced earlier in the methods section where we have justified its use with reference to several studies which have compared several calving laws (including running simulations into the future with various calving laws). Additional discussion of the choice of calving law has also been added into the Model Limitations section in the discussions.

We have run an additional simulation where we keep all forcings the same as the spin-up. However, we ran the spin-up to match historical mass loss as well as we could (and believe our mass loss trend of 0.9 Gt/yr compares well to the mass loss trend between 2000 and 2017 of 0.96 Gt/yr from Mouginot et al (2019)). This is in contrast to ISMIP6 where Goelzer et al state that '*In any case, representing the historical mass loss accurately was not a strong priority for our experimental set-up, where any background evolution is effectively removed by subtracting results of experiment ctrl_proj'*. In addition, Goelzer et al (2020) state that by subtracting the ctrl simulation SLR contribution from other results, committed sea level rise is not accounted for/included in the projections. One of the benefits of focusing on a single glacier is that we can spend the time matching our spin-up mass loss trends (as well as velocities/thicknesses etc) to observations and so run forward simulations where we don't have to neglect committed sea level rise. However, it is true that there is undoubtedly also some model drift in the results and that the Ctrl simulations provides valuable information on what the difference is between an SSP1-2.6 future and one where any climatic changes were to immediately cease. As such, we have added text relating our Ctrl experiments at several points throughout the results and discussion sections, as well as editing some figures some include this data. Overall, we find that c. 0.3 mm of SLR by 2300 is due to the combined effects of model drift and committed SLR.

We have also run simulations with no calving (one for low emissions, one for high emissions), edited figures 3 and 7 with these results, and added in some discussion of these results. In summary, no calving makes little difference to the high emissions simulations (retreat rates remain similar as the ice tongue/parts of the glacier just melt away from the negative SMB), but leads to an ice tongue which persists until 2300 under a low emissions scenario.

*Specific comments*
Line 1: how many glaciers with floating ice tongues still exist outside of northern Greenland? if none, change 'some of' to 'all'.
This really depends on whether one assumes Northern Greenland refers to the Northern sector of Greenland as defined by Mouginot et al (2019) and shown in Fig.1, or refers to e.g. the NE, NO, and NW sectors. We have changed the wording to make it clear that we just refer to the NO sector here, as we discuss this sector later in the text.

Line 5: suggest changing 'increased mass loss from discharge' to just 'increased ice discharge'

Changed

Line 33: State some references of studies that have presented these 'observational records'.
References have been added

Line 40: Throughout I would clearly state what you mean by 'stable' i.e. 'mass balance is close to zero' or 'limited change in front position'
Clarification on what we mean by stable has been added to the relevant sentences

Figure 1: See comment above about Section 2. I think this study figure should focus on Ryder Glacier and there is no need to necessarily include the others. Also the inset currently covers half of Ryder Glacier's catchment. Possibly show the model domain/mesh in this figure and the flowline used in other figures. I think the colour bar should be labelled speed rather than velocity.

We have replaced figure 1 with a figure focusing on Ryder Glacier, with a panel where the entire model domain can be seen.

Line 116: Heading 'Numerical model' doesn't really make sense given that this section includes a large part on input data. Perhaps consider changing to Methods and then 3.2 could be 'Model-set up' or similar.

The entire modelling methods section has been re-worked, including changing the section heading to 'Methods' as well as making changes to several of the subsection headings

Line 120: Does the domain allow for advance during the relaxation?
It does, and we have added extra clarification in here

Line 136: State the resolution of the velocity dataset.
We have added in that the resolution of the dataset is 250m

Line 138: I am not totally convinced by this justification for using Budd friction law, is there any reason to suggest that because it works best for Petermann it would for Ryder? I think if not testing different sliding laws in this study, the potential impact of this choice needs to be in the Discussion, especially given that Akesson et al. 2021 show this can have a large impact on future projections at Petermann.
We chose a Budd law in part due it having been found to work well for Petermann, but also as this sliding law has been used for a recent study looking at the whole of Greenland (Choi et al., 2021). As such, we are then able to easily compare our results to the results of these studies. We agree that sliding choice can have an impact on results, but note that other recent studies found other factors to be more important. For example, Barnes and Gudmunsson (2022) found that model results show similar decade-to-century scale predictive power regardless of sliding law choice whilst Carr et al (2024) found sliding law choice to far less import than e.g. SMB. As this study operates on the timescales identified as robust by Barness and Gudmunsson, and focuses on factors such as SMB and ocean forcing, we believe the results to be valid without testing additional sliding laws. We have added extra discussion of this into the 'Model Limitations' section.

Line 145: Is it common to have a spatially uniform viscosity field over the grounded ice?

Perhaps mention some other studies that have used this approach. It may also warrant a space in the limitations section of the Discussion.
This approach is relatively common – it is the approach taken in the aforementioned Åkesson et al (2021) and Choi et al (2021) studies, as well as additional studies such as Wilner et al (2023), Humbert et al (2023), and Åkesson et al (2022). We have added all of these references.

Lines 146-156: I think these lines would be better placed in a section 'Relaxation' because it is currently confusing to discuss this before introducing the calving parameterisation Eq. 2 and introducing the SMB forcing in Section 3.2.1.

We have reshuffled the entire methods section, including the creation of a 'Model Relaxation' subsection as suggested

Figure 2: If possible make panel d bigger so it's possible to see some of the detail in the change through time, although I appreciate the geometry is not changing much during this relaxation.

Panel d has been made bigger (stretched the y-axis)

Line 166: More details on these SSP forcing are needed. Which CMIP model was used? Are these an ensemble mean? Also do you apply these as anomalies ontop of the mean SMB field used during the relaxation?
We have specified that the SMB fields are the mean from CMIP6 and that there are independent of the historical SMB field (e.g. not applied as anomalies).

Line 176: What does a 'high level of subglacial discharge' equate to?
We have added in that a high level of subglacial discharge is set to equal 3.88 km3 yr-1 as defined by the Wiskandt et al (2023) study

Line 199: I'm not sure what you mean by 'Ryder Glacier's grounded calving front during the relax- ation', surely the calving front is floating during the relaxation? make this clearer to the reader.
Ryder Glacier currently has two calving fronts; a large floating front and a smaller, grounded front. We have edited the text to state this clearly in the introduction when we first mention Ryder, as well as changing the text in the methods so that readers are reminded of this.

Line 206: How was the value of σmax varied throughout the simulation? linearly reduced each year?
Yes, it was linear and we have added this in

Line 228: when you say 'grounded fronts' do you mean when the ice fronts become grounded during the experiment when the ice tongue is lost? I think a better explanation of what you mean by grounded fronts throughout would be useful.

We have clarified that this applies to any grounded part of the terminus. In combination with the additional text clarifying that Ryder already has a grounded terminus, we hope that the situation(s) in which this is applied are clarified.

Figures 4 and 5: units for thickness change.

This has been added in both figures

Figure 6: Where is this frontal velocitiy taken from? Is this the most useful measure? I would have instead reccomended to show the change in velocity inland of the terminus, to show how changes at the front (thinning/calving) impact inland grounded ice flow.

We have changed this so that the purple lines now show velocity inland of the terminus. We take the mean velocity of all grounded ice areas within 500m of the grounding line, evaluated for each time step.

Figure 7: This Figure needs a bit of a rethink, the lines in panels a, b, i, j are impossible to see any trend in and are too small. I think the sign of the red line in b) is wrong if you are expressing discharge as a loss.

The main idea behind the line graphs was to show the overall trend of mass loss in different simulations, for example by highlighting how much mass loss accelerated after 2100 in the high emissions scenarios/ how much discharge increased once low emissions simulations reached a bedrock depression. Unfortunately, the visuals of these very large increases meant that the y axis had to cover a large range of values, making the lines sometimes hard to see. Although we had tried to ameliorate this issue by showing a zoom in of the lines on panel a, we can understand how the lines remained hard to distinguish. As such, we have removed all the line graphs from this figure and just kept the pie charts.

Section 4.3: was the integrated SMB calculated over the grounded area only? and how was discharge calculated? across the grounding line? or a defined flux gate?

The integrated SMB across the model domain at each timestep is available as an output from the model. In additional, the total mass loss at each timestep is available as an output. As such, the discharge can then be calculated as the difference between these two. We have added in information on how the discharge was calculated.

Line 286: I find 'discharge losses' to be a bit awkward to read, I suggest changing it throughout to 'increases or decreases in ice discharge'.

Changed

Line 298: Surely the fact that the ice tongue was lost quite quickly in the high-end simulations is one of the reasons for the submarine melt rates not having such an impact in these simulations. Perhaps worth mentioning.

After the ice tongue is lost, we still have submarine melt; it is just applied to the grounded front instead. As overall discharge still remains higher in the high emissions scenarios (and continues to grow with time), the lack of sensitivity to ocean scenario is more likely due to the fact that the SMB just becomes so negative that it dwarfes discharge (see Fig. 7, where the high and increasing levels of discharge in the HE simulations can be seen in the pie charts). We have, however, added in some information on how the overall magnitude of submarine melt changes upon ice tongue loss.

Lines 302-315: All of this could be shortened and summarised in a few sentences. Also if

making a comparison between your study and other Greenland wide results I suggest including the ISMIP6 results.

We have removed some content from this section to streamline the discussions, but also added in several sentences comparing our results to Goelzer et al (2020).

Line 321: See comment above, surely the reason there is less sensitivity to ocean forcing is because at some point there is no ice tongue left? Worth mentioning this.

See response above

Line 337: The impact of the melt-elevation feedback may not be straightforward. Consider adding a reference to Delhasse et al., 2024 - TC who showed in coupled simulations the positive melt-elevation feedback may be mitigated.

We have added in a sentence addressing this point, along with a reference to the Delhasse et al study

Line 380: 'total ice-tongue collapse' to me suggests instantaneous collapse, i.e. the timeframe of C.H. Ostenfeld ice tongue collapse, whereas as far as I understand in your experiments the removal of the ice tongue is gradual due to the calving law? in which case I'm not sure have replicated an entire/immediate tongue collapse and it would be worth discussing this.

This is true – we have changed the word 'collapse' to 'loss' to remove the connotation that the loss of the ice tongue was sudden

Line 392: see comment above, I suggest using 'increased ice discharge' instead of 'discharge losses' throughout.

Fixed.

Lines 426-432: These lines are an exact duplicate of Lines 377-382. I suggest a careful proof read of the entire Discussion making sure there is no duplication and that the information presented is as concise as possible.

We have removed the first instance of this duplication, thank you for noticing this.

Line 455-460: This section on topography is important, but I do wonder about mentioning that loss of the ice tongue doesn't appear to initiate a runaway retreat of the grounding line further inland.

True – we have added a sentence that makes it clear that topography/bathymetry can also stabilise the glacier (something we see in the simulations once a prograde slope is reached).

Section 5.4: I think the limitations on choosing a single sliding law and calving law need to be discussed in more depth here.

More discussion of the sliding law has been added in (see response further up to your initial comment on the sliding law). The 'model limitations' section already contains discussion of how calving characteristics may change in the future but we have now additionally included a line specifying that different calving laws would be beneficial. However, it is not possible within the computation time allocated to run all of the simulations with several different sliding and/or calving laws and we believe that choosing a calving law based off recent comprehensive comparisons is a valid decision.

Line 505: This sentence would benefit from rephrasing. State the pathway instead of using 'latter'. 'greatly reduced negative societal impact' reads awkwardly as well

This sentence has been rephrased.

**Reviewer 2:**

*Summary:*
In this study, the authors modeled Ryder Glacier in northern Greenland under different climate scenarios to assess the risk and consequences of ice tongue loss. The model results suggested that ice discharge is the primary contributor to mass loss until 2100 under both low and high emission scenarios assumed in the study. After 2100, the results show that surface mass balance becomes the dominant factor driving mass loss under the high emission scenario until 2300, while ice discharge remains the largest contributor under the low emission scenario. The authors compared their results with observations and other modeling studies for neighboring glaciers, Petermann and C.H. Ostenfeld, and highlighted the importance of topographical controls on heterogenous behaviour in neighbouring glaciers.

Overall, this study raises several interesting discussion points and provides valuable insights for Ryder glacier, which, as noted in the manuscript, has not been studied as extensively as other glacier in northern Greenland. These findings could also provide valuable insights for other glaciers in northern Greenland and highlight the importance of high-resolution modeling of individual glaciers to better assess the future change in individual glaciers.

However, the manuscript needs revisions, including improvements to the modeling framework part and the overall structure of the writing. Below are the major and minor comments that I believe should be address before publication.

Thank you for your constructive review of the manuscript. Our responses to all of your comments are detailed below.

*Major comments:*
*1. Model relaxation.*
The model relaxation part (L146-156) is somewhat confusing as it lacks details about the calving parameterization which is only introduced later in section 3.2.2. In addition, I would appreciate more results and clarity regarding this model relaxation. For example, is the purpose of the relaxation to match current observations? How closely does the modeled melt rate match to published present-day melt rate (L149-151)? Is it spatial pattern or the maximum melt rate that you try to compare? How does the mass loss change during this relaxation period compared to the observation? How does the changes in ice front positions compare to observations during the relaxation period? Providing these details would improve the modeling part.

Thank you for this comment, something which was also brought up by the other Reviewer. The entire methods section has been reshuffled with the aim of more clearly describing the model set-up and relaxation. We have created a new subsection 'Model relaxation' where we have more clearly stated the aims of the relaxation and explicitly compared our relaxed model with several metrics. Some of these comparisons were also in the previous version of the manuscript (e.g. mass loss during the relaxation was compared with observations in lines

L151-153), but hopefully this is presented with more clarity in the new structure. The basal melt parameterisation is described in more detail, alongside additional references to the observational data, in the new subsection 'Model set-up'.

**2. Manuscript structure**

Consider revising the structure of 1. Introduction and 2. Petermann, Ryder, and C.H. Ostenfeld Glaciers section. It seems that the primary research questions the authors aim to address in this study are outlined in Section 2 (L107-L109), rather than at the end of the introduction (L44-47). These different questions may confuse readers about the main objectives of the study. Additionally, the comparison between three neighboring glaciers is too detailed in section 2, which may mislead/confuse the readers on the focus of this study. Consider summarizing this comparison part and rephrasing it to emphasize Ryder glacier's unique characteristics and the study's goals.

In addition, I think section 3.1. is unnecessary and could be merged with other modeling descriptions. Consider rewriting the modeling part as well, particularly the model relaxation part and related forcings.

Thanks for this comment and related suggestions – similar issues were also raised by the other Reviewer. We have removed section 2 ('Petermann, Ryder and C.H. Ostenfeld glaciers'), and replaced the study area figure with one focusing on Ryder glacier. Some material from this now-removed section has been incorporated into the introduction or the discussion section 'Comparison with other Glacier-Fjord systems'.

The section 'Topographic data' (prev. 3.1) has been removed, with the content merged into the broader 'Methods' section. The 'Methods' section detailing the model inputs and experimental design has been reworked into a new structure which now includes a specific 'Model Relaxation' subsection.

**Specific comments:**

L18: … are already higher that at any time during … -> … are already higher than at any time during…

Changed.

L27: …that this this sector contains… -> that this sector contains…

Changed.

L33: "Observational records from recent decades have shown a large degree of spatial…surrounding the future behaviour of glaciers". Add references to support this statement.

References have been added in (Porter et al., 2014; 2018; Cooper et al., 2022)

L53-55: The mean SMB per area is compared between the three glaciers, but what about the spatial pattern of SMB? For instance, are there more negative SMB values near the front of Ryder Glacier compared to the other glaciers? Highlighting such patterns could strengthen the argument.

This data is from the supplementary material of Mouginot et al (2019), where the SMB is given as an integrated value for each drainage basin. However, as part of the re-structuring, this sentence no longer exists.

L61-62: This sentence is unclear. Are you comparing the highest melt rates at Ryder Glacier to the overall melt rates at Petermann Glacier? Clarify the comparison.

This sentence has been changed to: '*Previous observational records of basal melt show higher overall melt fluxes at Petermann Glacier than Ryder Glacier, but with the Eastern side of Ryder Glacier's grounding line experiencing melt rates comparable to or even higher than Petermann Glacier at around 60 m yr-1 (Wilson et al., 2017).*'

L64: "higher grounding line melt rates than what?" Clarify the reference point for comparison.

It has been clarified that the comparison is between Ryder and Petermann Glaciers

L85: "The lack of lateral friction at C. H. Ostenfeld's ice tougue…survived to the present day" Add references.

References added

L120: Is the model domain the same as the green basin shown in Fig.1? If so, clarify this in the text.

This has been clarified, and the domain is more clearly seen in the new Fig.1 which focuses solely on Ryder Glacier.

L125: "performing considerably better" Does this mean more accurate? Better in terms of what specific metric or criteria?

We had clarified that the misfit between HO and FS simulations is smaller than between SSA and FS simulations

L128: "…satisfy the Courant-Friedrichs-Lewy condition". Add references.

Reference added.

L174: "…increase by around 1oC by 2100 and around 2oC by 2200". What climate scenario are these values based on? Why not using the same future climate scenarios (SSPs) as the SMB forcing? This needs to be justified.

We have added the following into the text:
1. L173: 'Subsurface ocean temperatures around North Greenland are projected to increase by around 1∘C by 2100 and around 2∘C by 2200 (Yin et al., 2011) under a mid-range increase in greenhouse gas emissions. These numbers were chosen as they are based on an ensemble of model runs, are specific to North Greenland, and target increases in the subsurface water temperatures thought to be of importance for Ryder Glacier (Jakobsson et al., 2020).'

2. L189: 'Although the magnitude of ocean warming as well as the magnitude of subglacial discharge is likely to vary depending on future emissions scenario, by running the same suite of ocean scenarios for both our low and high emissions simulations, we can compare the sensitivity of Ryder Glacier to ocean forcing under different SMB-scenarios.'

L191: Add some transition sentences before explaining calving parametrization to improve the flow.
The following has been added: 'Calving is another important mass loss process at Ryder Glacier's marine margin, with the choice of calving law being identified as a key source of uncertainty in models of Greenland's future (Goelzer et al., 2020)'

L196: "…best match observations of front position…": Specify the year or period of the observations you are referring to.

We have specified that we match the front position to the overall trend of 'no movement' since 2000.

L199: "Finally, the stress threshold for grounded ice…during the relaxation": consider rewriting this sentence. Isn't the calving front floating during the relaxation? Please clarify it.

Ryder Glacier presently has two termini; the (main and much larger) floating terminus, and a second grounded terminus. We have specified this more clearly in the text where Ryder Glacier is first described as well as changing the wording in this section.

Introduction: '*Petermann and Ryder glaciers are two of the few Greenlandic glaciers which still have a floating ice tongue, although Ryder Glacier also has a smaller, grounded terminus*'

Methods: '*... this led to the best match with Ryder Glacier's second, smaller grounded calving front…*'

L205: reducing the value linearly in time?

Yes, it was a linear reduction – something we have now specified

L228: "In all simulations, the melt rate applied across the … below the floating tongue" : Was this frontal melt rate also applied during the relaxation simulation?

Yes, and we have clarified this into the text

L287: "In a high emission scenario…" Do you mean all high emission scenarios? or is this the mean value from multiple high emission scenarios?

This statement is true for all high emissions scenarios, and we have edited the text to clarify this ('In a high emissions scenario' to 'In all high emissions scenarios')

L360: "show acceleration occurring" -> "show acceleration of retreat?"

We have changed this sentence as it was badly worded. It now reads: *'Instead, both the simulations shown in Fig. 6 show acceleration between 2080 and 2150 as the ice tongue shrinks, with a deceleration then being seen post ice tongue loss - something that coincides with an acceleration of SMB-driven thinning.'*

---

## Referee Report (RR1)

**Review of Holmes et al. 'Sea-level rise contribution from Ryder Glacier in Northern Greenland varies by an order of magnitude by 2300 depending on future emissions'**

**General comments**

This is my second review of the manuscript by Holmes et al. which presents numerical modelling simulations of Ryder Glacier to project sea level rise contribution by the year 2300. The manuscript is substantially improved over the previous version and I greatly appreciate the authors efforts to address my comments, in particular improving the abstract, introduction, and including a more thorough presentation of the model relaxation and control simulations. On a second read through I have a few remaining line comments prior to publication that are detailed below.

**Specific comments**

**Line 6:** change "to Ryder Glacier" to "of Ryder Glacier"

**Line 8:** "degree" would be better with a temperature, suggest changing to "amount of ocean warming"

**Line 17:** Suggest removing second instance of "current" in this sentence

**Line 25:** This sentence would be more concise as "...uncertainties on the future behaviour of this region are considered larger than other sectors of the ice sheet"

**Line 40:** It's slightly confusing to the reader that you say "reasonably stable" here and then in a couple of lines say it has "periods of both advance and retreat". Suggest rephrasing here, to make it clear there has been little net retreat over the past 15 years.

**Line 52:** I think "knock-on impacts" could be clearer, or just removed to be "and the potential impact on sea level rise projections"

**Line 57:** Consider adding a reference to Rückamp et al. 2019: *Calving Induced Speedup of Petermann Glacier*

**Line 67:** Suggest removing "rather puzzling", it is a bit of a vague statement

**Line 76:** I suggest labelling Sherard Osborn fjord on Figure 1 so the reader knows where it is.

**Line 130:** You state "CESM2" and "RACMO" datasets, but do not mention these again when you introduce the specific datasets. In the rest of the methods make it clear where you use SMB from RACMO and CESM2 (e.g. line 139). I also think this sentence would be better at the start of this paragraph, when you first introduce SMB forcing in the model. Then go onto detailing the SMB-elevation feedback.

**Line 243:** Heading might be better as "Terminus" retreat.

**Line 246:** Change "spin-up" to "relaxation", spin-up hasn't been used anywhere in the text before.

**Figure 7:** This figure is much improved and easier to interpret. I suggest making the circles a bit bigger and making the text (especially the SMB Gt/yr) less cramped to improve readability. The legend font could be smaller.

**Line 324:** The statement here "Ryder Glacier is not expected to follow the same trajectory as the

Greenland ice sheet as a whole" makes me wonder why you have compared the exact values for your simulations to those of previous Greenland wide studies in the earlier part of this paragraph. It is of course interesting to compare the impact of atmospheric versus oceanic forcing and SMB versus discharge between your study and Greenland wide ones, e.g. ISMIP6, but direct comparison of mm SLE might be less useful.

**Line 359:** Be careful here, marine-ice sheet instability is not just acceleration and thinning over a retrograde slope, but rather than this ice loss is irreversible. I would rephrase to something along the signs of "may be an indication of marine ice sheet instability"

**Line 378:** "This trend of an acceleration" reads a bit awkwardly, suggest rephrasing.

**Lines 447-455:** I was also expecting a statement in this paragraph about how the bathymetric sill in Ryder fjord prevents warm water intrusion, perhaps you could include this and add something about how you think this may play a role in future projections. - update - I see this statement about the sill appears further down in the model limitations section, but I'm not sure it is a model limitation really, just a potential control on melt rates. I would suggest moving into Section 4.3 (here).

---

## Referee Report (RR2)

**Review:**

I would like to thank the authors for revising the manuscript in response to the reviewers' comments. The revisions have significantly improved the clarity and overall quality of the manuscript. At this point, I have only a few minor suggestions, which are described below. I recommend the publication of this manuscript in TC following minor revisions.

**Specific comments:**

L104: *"… so that is varies linearly…"* => …so that **it** varies linearly…

L111: *"…set to equal half of the maximum melt rate below the floating tongue.":* Why is it set to half of the maximum basal melt rate? Please clarify and add a reference.

L177-184: This part reads more like a discussion. Consider moving this paragraph to the Discussion section.

L248: *"Here, …"* => "In the Ctrl simulation, …"

L283: *"the spike in frontal velocities up to 900 m/yr…"*: Isn't the spike in the mean frontal velocity 2,000 m/yr in Fig. 6a?

L290: (Fig. 7b) => (Fig. 7f)

Figure 6 (caption): "Where the pink and green lines meet, …" => "Where the **magenta** an green lines meat,.."

---

## Author Response (AR2)

**Thank you to both reviewers for taking the time to re-review our manuscript, and for your positive comments about the edits we have made. We have detailed our responses to the new set of reviews below.**

**Yours sincerely,**

**Felicity Holmes (on behalf of all co-authors)**

**Reviewer 1:**

**General comments**

This is my second review of the manuscript by Holmes et al. which presents numerical modelling simulations of Ryder Glacier to project sea level rise contribution by the year 2300. The manuscript is substantially improved over the previous version and I greatly appreciate the authors efforts to address my comments, in particular improving the abstract, introduction, and including a more thorough presentation of the model relaxation and control simulations. On a second read through I have a few remaining line comments prior to publication that are detailed below.

Thanks for your positive comments about the changes we have made since the last review.

**Specific comments**

Line 6: change "to Ryder Glacier" to "of Ryder Glacier"

Changed

Line 8: "degree" would be better with a temperature, suggest changing to "amount of ocean warming"

Changed

Line 17: Suggest removing second instance of "current" in this sentence

We have removed the first instance of 'current'

Line 25: This sentence would be more concise as "...uncertainties on the future behaviour of this region are considered larger than other sectors of the ice sheet"

We have changed this sentence in accordance with your suggestion

Line 40: It's slightly confusing to the reader that you say "reasonably stable" here and then in a couple of lines say it has "periods of both advance and retreat". Suggest rephrasing here, to make it clear there has been little net retreat over the past 15 years.

We have changed 'remained reasonably stable' to 'shown little net movement'

Line 52: I think "knock-on impacts" could be clearer, or just removed to be "and the potential impact on sea level rise projections"

We have changed 'knock-on impacts' to 'and the potential impact on sea level rise projections'

Line 57: Consider adding a reference to Ruckamp et al. 2019: Calving Induced Speedup of Petermann Glacier

The reference has been added in

Line 67: Suggest removing "rather puzzling", it is a bit of a vague statement

We have removed this phrase

Line 76: I suggest labelling Sherard Osborn fjord on Figure 1 so the reader knows where it is.

Sherard Osborn fjord has been labelled in Fig.1a

Line 130: You state "CESM2" and "RACMO" datasets, but do not mention these again when you introduce the specific datasets. In the rest of the methods make it clear where you use SMB from RACMO and CESM2 (e.g. line 139). I also think this sentence would be better at the start of this paragraph, when you first introduce SMB forcing in the model. Then go onto detailing the SMB- elevation feedback.

The reference datasets are now introduced before we talk about the SMB-elevation feedback. We have added in e.g. that the historical SMB was from RACMO/ the future datasets are CMIP6 forced RACMO when they are specifically introduced later in the methods.

Line 243: Heading might be better as "Terminus" retreat.

The heading, as well as the first sentence of the section, has been changed so that 'Margin' is now called 'terminus'

Line 246: Change "spin-up" to "relaxation", spin-up hasn't been used anywhere in the text before.

Changed

Figure 7: This figure is much improved and easier to interpret. I suggest making the circles a bit bigger and making the text (especially the SMB Gt/yr) less cramped to improve readability. The legend font could be smaller.

The circles are bigger, more room is given for the text, and the legend font size has been decreased.

Line 324: The statement here "Ryder Glacier is not expected to follow the same trajectory as the Greenland ice sheet as a whole" makes me wonder why you have compared the exact values for your simulations to those of previous Greenland wide studies in the earlier part of this paragraph. It is of course interesting to compare the impact of atmospheric versus

oceanic forcing and SMB versus discharge between your study and Greenland wide ones, e.g. ISMIP6, but direct comparison of mm SLE might be less useful.

We have added a sentence clarifying that we cannot make more targeted comparisons due to a lack of previous studies focusing on Ryder, hence explaining why we are forced to compare our results for Ryder to results from studies focusing on the entire ice sheet

Line 359: Be careful here, marine-ice sheet instability is not just acceleration and thinning over a retrograde slope, but rather than this ice loss is irreversible. I would rephrase to something along the signs of "may be an indication of marine ice sheet instability"

We have changed this sentence to '..highlighting the strong coupling between bedrock topography and mass loss'

Line 378: "This trend of an acceleration" reads a bit awkwardly, suggest rephrasing.

We have rephrased this sentence (removed 'trend of an')

Lines 447-455: I was also expecting a statement in this paragraph about how the bathymetric sill in Ryder fjord prevents warm water intrusion, perhaps you could include this and add something about how you think this may play a role in future projections. - update - I see this statement about the sill appears further down in the model limitations section, but I'm not sure it is a model limitation really, just a potential control on melt rates. I would suggest moving into Section 4.3 (here).

We have moved this discussion point to sect. 4.3

**Reviewer 2:**

I would like to thank the authors for revising the manuscript in response to the reviewers' comments. The revisions have significantly improved the clarity and overall quality of the manuscript. At this point, I have only a few minor suggestions, which are described below. I recommend the publication of this manuscript in TC following minor revisions.

Thank you for your comments on our revised manuscript

**Specific comments:**

L104: "… so that is varies linearly…" => …so that it varies linearly…

Changed

L111: "…set to equal half of the maximum melt rate below the floating tongue.": Why is it set to half of the maximum basal melt rate? Please clarify and add a reference.

We have added in clarification of this point – that the melt rate for grounded fronts was set to correspond to recent behaviour in the absence of any recorded melt rates (or ocean temperatures) for this terminus.

L177-184: This part reads more like a discussion. Consider moving this paragraph to the Discussion section.

We have moved part of this section to the discussion and left a shortened version.

L248: "Here, …" => "In the Ctrl simulation, …"

Changed

L283: "the spike in frontal velocities up to 900 m/yr…": Isn't the spike in the mean frontal velocity 2,000 m/yr in Fig. 6a?

Changed

L290: (Fig. 7b) => (Fig. 7f)

Changed

Figure 6 (caption): "Where the pink and green lines meet, …" => "Where the magenta an green lines meat,.."

Changed

---

## Author Response (AR3)

2025-04-29

Felicity Holmes
Department of Geological Sciences
Stockholm University
106 91 Stockholm
felicity.holmes@geo.su.se

**Author's response**

**Dear Dr. McCormack,**

We appreciate your time in handing our manuscript submission, and for providing helpful comments which have improved the manuscript.

Please see below for our responses to your requests. Your comments are shown in bold, with our responses given in normal typeface.

Thank you very much for your time,

Yours Sincerely,

Felicity Holmes, on behalf of all authors

**- Abstract Line 1 (and elsewhere throughout manuscript): Greenland ice sheet → Greenland Ice Sheet**

Changed in all cases

**- Line 67 (and elsewhere?): "in light of its current stability". Recommend to avoid using the word stability unless referring to specific, mathematical stability. Perhaps use "persistent grounding line" or "limited recent changes" instead**

Changed on line 67, as well as in several other locations (e.g. lines 45 and 457)

**- Line 437: delete repeated "response"**

Changed